# Assessing the effects of stress on feeding behaviors in laboratory mice

Marie Francois[1], Isabella Canal Delgado[1], Nikolay Shargorodsky[1], Cheng-Shiun Leu[1,2], Lori Zeltser[1,3]*

[1]Naomi Berrie Diabetes Center, Division of Molecular Genetics, Columbia University Irving Medical Center, New York, United States; [2]Department of Biostatistics, Columbia University Irving Medical Center, New York, United States; [3]Department of Pathology and Cell Biology, Columbia University Irving Medical Center, New York, United States

**Abstract** Stress often affects eating behaviors, increasing caloric intake in some individuals and decreasing it in others. The determinants of feeding responses to stress are unknown, in part because this issue is rarely studied in rodents. We focused our efforts on the novelty-suppressed feeding (NSF) assay, which uses latency to eat as readout of anxiety-like behavior, but rarely assesses feeding per se. We explored how key variables in experimental paradigms – estrous and diurnal cyclicity, age and duration of social isolation, prandial state, diet palatability, and elevated body weight – influence stress-induced anxiety-like behavior and food intake in male and female C57BL/6J mice. Latency to eat in the novel environment is increased in both sexes across most of the conditions tested, while effects on caloric intake are variable. In the common NSF assay (i.e., lean mice in the light cycle), sex-specific effects of the length of social isolation, and not estrous cyclicity, are the main source of variability. Under conditions that are more physiologically relevant for humans (i.e., overweight mice in the active phase), the novel stress now elicits robust hyperphagia in both sexes . This novel model of stress eating can be used to identify underlying neuroendocrine and neuronal substrates. Moreover, these studies can serve as a framework to integrate cross-disciplinary studies of anxiety and feeding related behaviors in rodents.

## Editor's evaluation

All of the reviewers have found your response to the previous critiques thoughtful and thorough. Your studies will serve as a valuable resource to the field and will be widely cited.

## Introduction

Most studies in humans focus on stress-induced overeating (also called emotional eating or stress eating), but stress can also lead to decreased caloric intake in some people (*Greeno and Wing, 1994*; *Oliver and Wardle, 1999*; *Wallis and Hetherington, 2009*). Although stress-related eating behaviors are heterogeneous at the population level, individual behaviors are highly predictable (*Oliver and Wardle, 1999*; *Stone and Brownell, 1994*). Clinical and epidemiological studies support the idea that properties of the stressor influence the direction of the response. The effect of a mild stress is variable, but as the intensity of stress increases, people are more likely to eat less (*Stone and Brownell, 1994*; *Kandiah et al., 2008*). The type of stress also matters, with physical stressors more likely to suppress intake than psychosocial stressors (*O'Connor et al., 2008*). While stress affects eating in both men and women (*Rutters et al., 2009*), there are sex differences in responsiveness. Females are typically more sensitive to interpersonal and emotional stress, while males are more sensitive

*For correspondence:
lz146@cumc.columbia.edu

**Competing interest:** The authors declare that no competing interests exist.

**eLife digest** In times of heightened anxiety – say, during a global pandemic – many of us will reach for donuts or a particularly appetizing pizza for comfort. Others, however, will tend to shun food. What underlies these differences, and, in fact, the neural and hormonal pathways at play during stress eating (when people eat without being hungry due to emotional reasons), remain unclear.

This is partly because scientists lack good animal models in which to study these behaviors. In particular, female rodents are usually excluded from studies under the assumption that their hormonal cycles will disrupt the results. Yet, women are overrepresented in studies on feeding habits.

Modeling human behaviors using rodents is harder than it may appear. These animals are most active at night – yet most experiments are performed during the day. The same stressors also have different outcomes in males and females. François et al. therefore explored better ways to induce anxiety and evaluate feeding behavior in mice, hoping to reliably elicit stress eating.

The starting point was a common type of experiments known as novelty-suppressed feeding. First, mice are kept alone in a cage for up to two weeks on a normal diet so that they are used to experimental conditions. Then they are deprived of food overnight, before being given free access to food in the morning in a new environment. This stressful experience normally causes mice to take longer to eat than in their home cage. In rodents, the delay is thought to reflect stress as it is reliably reversed by anti-anxiety compounds approved for human use. In the novelty-suppressed feeding assay, both male and female animals exhibit signs of anxiety, but how much females eat is variable. François et al. showed that this variability is not due to hormonal changes, but instead to how long female mice had been kept alone.

Crucially, the test could be adapted so that mice would consistently exhibit behavior similar to human stress eating, whereby they eat more during the test without having fasted the night before. The changes included running the experiment at night, when the animals are normally most active, and using overweight mice (which captures the fact that, in humans, being overweight is associated with being prone to stress eating).

Stress eating is an important clinical issue, hindering weigh loss in people with obesity. The new model developed by François et al. could be adopted by other laboratories, enabling better research into this behavior.

to ego-threatening situations (*Tanofsky-Kraff et al., 2000*; *Laitinen et al., 2002*; *Clauss and Byrd-Craven, 2019*). Moreover, the threshold at which stress preferentially suppresses food intake is lower for males than females (*Stone and Brownell, 1994*).

While properties of the stressor shape eating behaviors, these influences cannot account for all of the heterogeneity. Even when a common stress is shared by many, such as the quarantine during the COVID-19 pandemic, both overeating and restricting are increased (*Phillipou et al., 2020*; *Coulthard et al., 2021*). These observations are consistent with the idea that physiological and psychological traits determine the direction of eating responses to stress at the individual level (*Stone and Brownell, 1994*). Elevated BMI is the variable most consistently associated with eating in the absence of hunger in both children (*Miller et al., 2019*) and adults (*Laitinen et al., 2002*; *Coulthard et al., 2021*; *Lemmens et al., 2011*). However, the underlying mechanism is unknown.

Basic principles governing stress-related eating behaviors are conserved in rodent models, providing construct validity. The likelihood of hypophagic responses increases with the stress intensity (*Levine and Morley, 1981*; *Michajlovskij et al., 1988*; *Martí et al., 1994*; *Harris et al., 1998*; *Vallès et al., 2000*; *Michel et al., 2005*; *Barfield et al., 2013*). Moreover, diet-induced obesity exaggerates hyperphagic behavior in models of social defeat stress in male mice and rats (*Bartolomucci et al., 2009*; *Razzoli et al., 2015*). Early experiments in rats reported stress-induced increases in food intake in males and females (*Antelman et al., 1976*). Over time, the field gradually shifted toward measurements of stress-induced suppression of eating behaviors as a readout of anxiety- or depressive-like states, without explicitly assessing caloric intake (*Cryan and Sweeney, 2011*; *Kokras and Dalla, 2014*). Current studies of stress-induced overeating are largely limited to models of binge-eating disorders, which involve restricted access to a palatable diet, often in combination with a chronic physical stress (*Boggiano and Chandler, 2006*). Rodent models recapitulate the preferential susceptibility

to binge-eating disorder and subclinical bingeing behavior in women (*Klump et al., 2011*; *Jacobi et al., 2004*; *Hudson et al., 2007*; *Croll et al., 2002*). However, there are no established models to study effects of short-term exposure to psychological stress on eating (reviewed in *François et al., 2021*).

There are several obstacles to studying the effects of acute stress on feeding behaviors in female rodents. They often exhibit reduced anxiety-like behaviors, especially when tasks involve locomotor activity or arousal (*Fernandes et al., 1999*; *File, 2001*; *Doremus et al., 2006*). Moreover, behavioral endpoints often vary across the ovarian cycle (*Becker et al., 2005*; *Beery and Zucker, 2011*). Together, these observations have been used as a rationale for excluding females in basic research. The paucity of rodent studies in females is particularly unfortunate, since the prevalence of anxiety, depressive symptoms, eating disorders, and subclinical disordered eating behaviors is higher in women (*Jacobi et al., 2004*; *Hudson et al., 2007*; *Altemus et al., 2014*). Moreover, single housing can exert opposite effects in males and females (*Oliver et al., 2020*), which is rarely considered when using this manipulation to measure feeding or other behaviors.

We set out to uncover aspects of experimental paradigms that promote hyperphagic vs. hypophagic responses to stress in female mice. We used the novelty-suppressed feeding (NSF) paradigm as the foundation for these studies. It has strong predictive validity with respect to antidepressant and anxiolytic therapeutics but is rarely exploited to examine the effect of stress on food intake (*Dulawa and Hen, 2005*). We performed 15 variations of the NSF assay to parse influences of sex, estrous and diurnal cyclicity, age and duration of social isolation, prandial state, diet palatability, and chronic high-fat diet (HFD) exposure on anxiety-like and feeding behaviors in mice.

## Results

### Estrous cyclicity is not the primary driver of heterogeneity in stress responses in females

We tested males and females in the manual NSF assay. We acclimated the mice to single housing for 2 weeks, subjected them to an overnight fast, and recorded responses to a chow diet in the home vs. novel cage environment (*Figure 1A*). Novel environment stress increased latency in both sexes (*Figure 1B*, males, Paired t-test: t=4.094; df=9; *p*=0.027 and females, Wilcoxon test: W=141; *p*=0.0002). Food intake was significantly decreased in the novel cage in males (*Figure 1E*, Wilcoxon test: W=-36; *p*=0.0078) but not in females (*Figure 1E*, Wilcoxon test: W=-67; *p*=0.0562). We examined the contribution of estrous cyclicity to the variability in the effects of novel environment stress. Females in diestrus (D) exhibited significantly higher latency (*Figure 1C*, One-way ANOVA: $F_{(3, 23)}$=22.52; *p*<0.0001) and lower food intake (*Figure 1F*, Kruskal-Wallis test: $H_{(3)}$=15.86; *p*=0.0012) in the home cage compared to males. In contrast, estrous cyclicity did not affect these behaviors in the novel cage (*Figure 1D, G*, Kruskal-Wallis test: $H_{(3)}$=16.80; *p*=0.0008 and One-way ANOVA: $F_{(3, 23)}$=2,370; *p*=0.0968, respectively).

Even though mice were acclimated to the experimental paradigm for 5 days, performing the NSF on the benchtop introduces stresses associated with an open cage and human contact. We examined the impact of minimizing these environmental stresses in females by performing the NSF assay in a system that allows automated control of food access and measurement of food intake. We evaluated adult females that were socially isolated for 2 weeks and provided with access to a chow diet in the morning after an overnight fast (*Figure 2A*). These minimally stressed conditions eliminated the effect of the estrous cycle (*Figure 2B, C, E, F*, Kruskal-Wallis test: $H_{(2)}$=1.269; *p*=0.5479, One-way ANOVA: $F_{(2, 18)}$=1.385; *p*=0.2757, One-way ANOVA: $F_{(2, 18)}$=0.0701; *p*=0.9326, and One-way ANOVA: $F_{(2, 18)}$=0.2336; *p*=0.7941, respectively). Latencies were lower in both the home cage ($X^2_{(1, n=38)}$ = 11.8, p = 0.001; *Supplementary file 1B-I*) and novel cage ($X^2_{(1, n=38)}$ = 22.2, p < 0.001; *Supplementary file 1B-I*), but the responsiveness to the novel environment stress was maintained (*Figure 2D*, Wilcoxon test: W=151; *p*=0.0071). Eliminating the effect of estrous cyclicity did not improve the consistency of the hypophagic response to novel environment stress (*Figure 2G*, Paired t-test: t=1.810; df=20; *p*=0.0854), arguing against the idea that estrous cycle is the primary driver of variability.

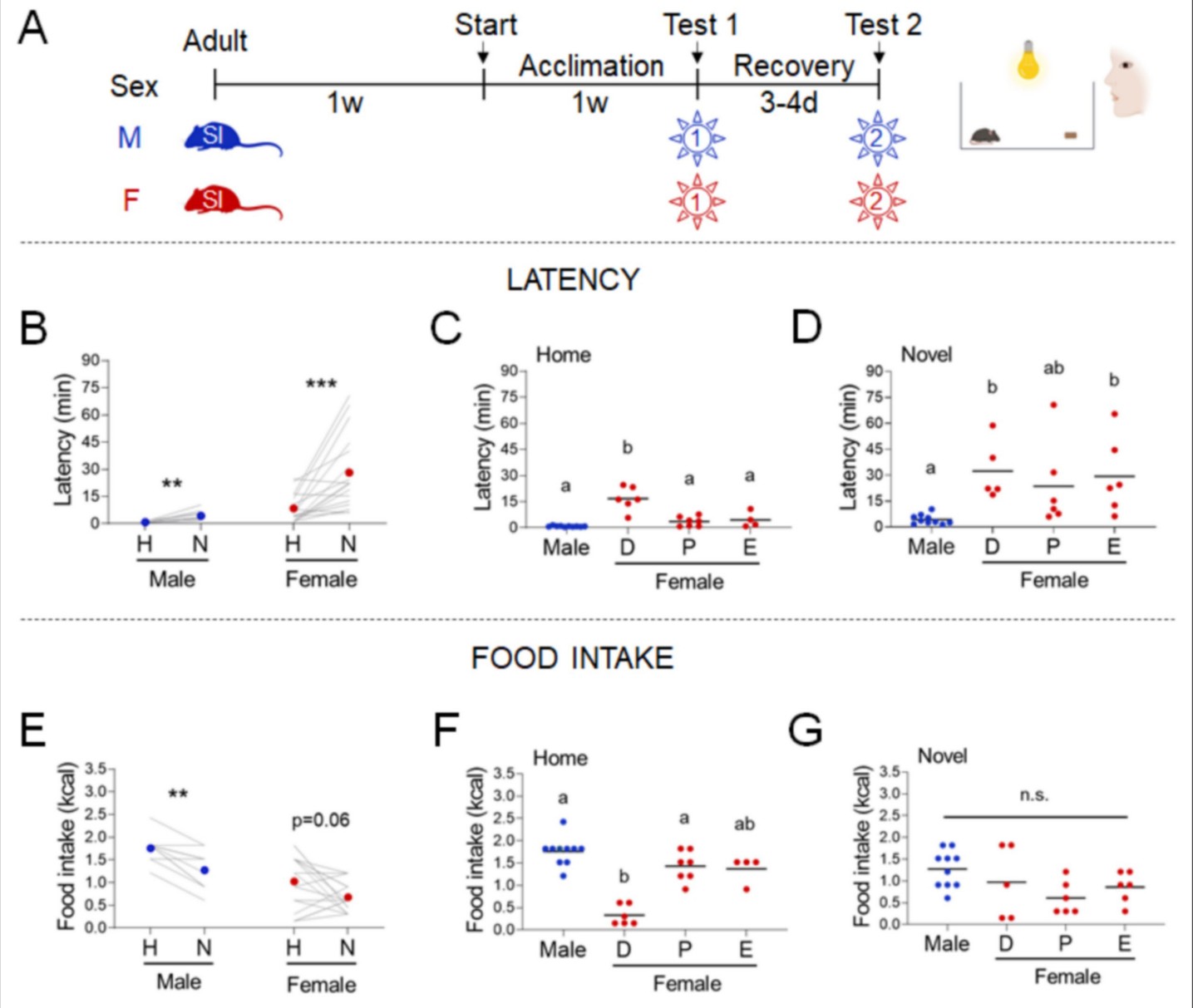

**Figure 1.** Effects of sex and estrous cyclicity in the standard novelty-suppressed feeding (NSF) assay. (**A**) Experimental paradigm for the standard NSF assay performed on the bench, in the morning following an overnight fast. (**B**) Latency to eat in home (H) and novel (N) tests in males (blue, $n = 10$, paired $t$-test: $t = 4.094$; $df = 9$; $p = 0.027$) and females (red, $n = 17$, Wilcoxon test: $W = 141$; $p = 0.0002$). (**C**) Latency to eat in the home test in males (blue, $n = 10$) and females (red, $n = 17$) categorized by estrous cycle stage (D: diestrus; P: proestrus; E: estrus) (one-way analysis of variance [ANOVA]: $F_{(3, 23)} = 22.52$; $p < 0.0001$). (**D**) Latency to eat in the novel test in males (blue, $n = 10$) and females (red, $n = 17$) categorized by estrous cycle stage (Kruskal–Wallis test: $H_{(3)} = 16.80$; $p = 0.0008$). (**E**) Food intake in home (H) and novel (N) tests in males (blue, $n = 10$, Wilcoxon test: $W = -36$; $p = 0.0078$) and in females (red, $n = 17$, Wilcoxon test: $W = -67$; $p = 0.0562$). (**F**) Food intake in males (blue, $n = 10$) and females (red, $n = 17$) categorized by estrous cycle stage in the home test (Kruskal–Wallis test: $H_{(3)} = 15.86$; $p = 0.0012$). (**G**) Food intake in males (blue, $n = 10$) and females (red, $n = 17$) categorized by estrous cycle stage in the novel test (one-way ANOVA: $F_{(3, 23)} = 2,370$; $p = 0.0968$). Significant differences denoted by different letters. **$p < 0.01$ and ***$p < 0.001$ between home and novel tests. SI, social isolation; n.s., not significant. See *Figure 1—source data 1*.

The online version of this article includes the following source data and figure supplement(s) for figure 1:

**Source data 1.** Standard novelty-suppressed feeding (NSF) assay in both sexes and across estrous cycle.

**Source data 2.** Sex hormones across estrous cycles.

**Figure supplement 1.** Evaluation of the estrous phase with cytology and hormone level measurements.

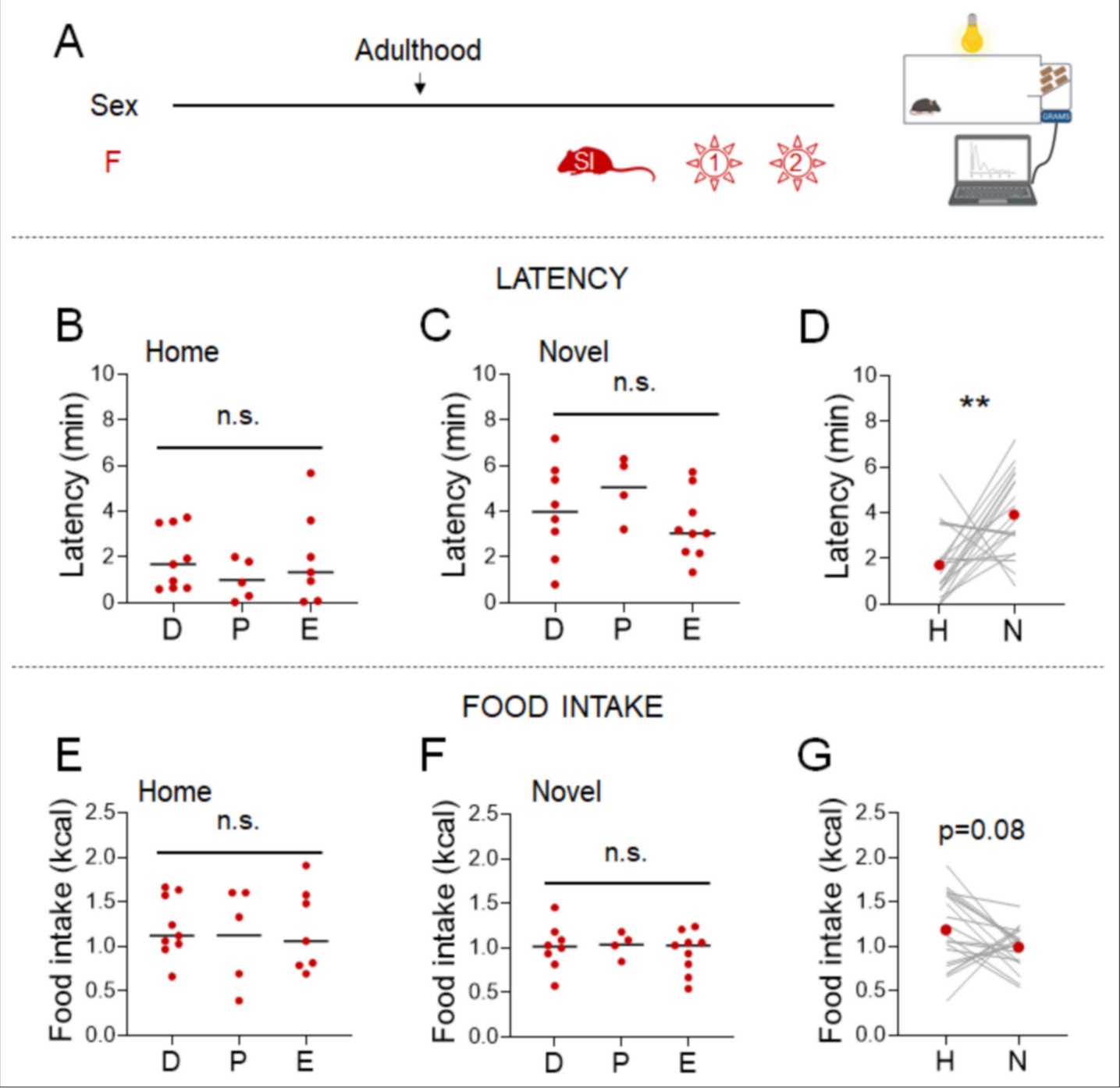

**Figure 2.** Effects of minimal environmental stress on the novelty-suppressed feeding (NSF) assay in females. (**A**) Experimental paradigm for the automated NSF assay performed in the automated recording system, in the morning following an overnight fast, in adult female mice socially isolated 2 weeks before the tests (*n* = 21). (**B**) Latency to eat in the home test in females categorized by estrous cycle stage (D: diestrus; P: proestrus; E: estrus) (Kruskal–Wallis test: $H_{(2)}$ = 1.269; p = 0.5479). (**C**) Latency to eat in the novel test in females categorized by estrous cycle stage (one-way analysis of variance [ANOVA]: $F_{(2, 18)}$ = 1.385; p = 0.2757). (**D**) Latency to eat in home (H) and novel (N) tests (Wilcoxon test: *W* = 151; p = 0.0071). (**E**) Food intake in the home test in females categorized by estrous cycle stage (one-way ANOVA: $F_{(2, 18)}$ = 0.0701; p = 0.9326). (**F**) Food intake in the novel test in females categorized by estrous cycle stage (one-way ANOVA: $F_{(2, 18)}$ = 0.2336; p = 0.7941). (**G**) Food intake in home (H) and novel (N) tests (paired *t*-test: *t* = 1.810; df = 20; p = 0.0854). **p < 0.01 between home and novel tests. SI, social isolation; n.s., not significant. See *Figure 2—source data 1*.

The online version of this article includes the following source data for figure 2:

**Source data 1.** Effect of minimal environmental stress on the standard novelty-suppressed feeding (NSF) assay in females.

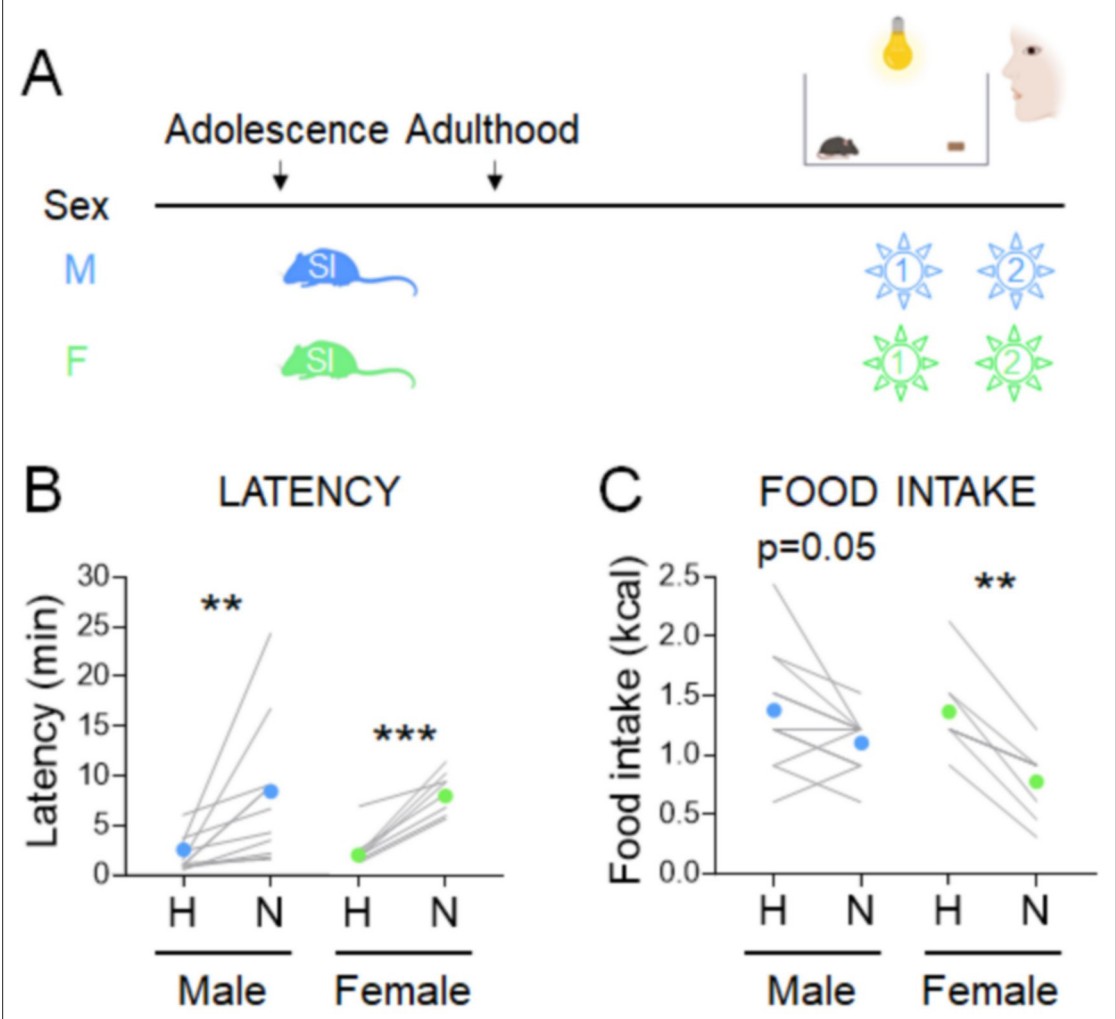

**Figure 3.** Effects of adolescent social isolation on the novelty-suppressed feeding (NSF) assay in males and females. (**A**) Experimental paradigm for the NSF assay performed in the morning following an overnight fast, in adult mice socially isolated at 5 weeks of age. (**B**) Latency to eat in home (H) and novel (N) tests in males (blue, *n* = 11, Wilcoxon test: *W* = 66; p = 0.001) and in females (green, *n* = 8, Wilcoxon test: *W* = 36; p = 0.0078). (**C**) Food intake in home (H) and novel (N) tests in males (blue, *n* = 11, paired *t*-test: *t* = 2.193; df = 10; p = 0.0531) and in females (green, *n* = 8, paired *t*-test: *t* = 6.347; df = 7; p = 0.0004). **p < 0.01, ***p < 0.001 between home and novel tests. SI, social isolation. See *Figure 3—source data 1*.

The online version of this article includes the following source data and figure supplement(s) for figure 3:

**Source data 1.** Effects of long social isolation starting at adolescence on the standard novelty-suppressed feeding (NSF) assay.

**Source data 2.** Parsing the effects of length vs. timing of social isolation on the standard novelty-suppressed feeding (NSF) assay in females.

**Figure supplement 1.** Parsing the effects of length vs. timing of social isolation on the standard novelty suppressed feeding (NSF) assay in females.

### Social isolation shapes stress responses in an age- and sex-dependent manner

We next examined another aspect of the NSF paradigm that could affect food intake in females – the 2-week period of social isolation stress needed to acclimate mice to the testing conditions. Timing of social isolation modulates anxiety-related behavior in a sex-specific manner (*Donner and Lowry, 2013*; *Walker et al., 2019*). We assessed the impacts of starting social isolation in adolescence (5 weeks) on behaviors in the manual NSF assay in adulthood in males and females (*Figure 3A*). All mice exhibited higher latencies in the novel cage, regardless of sex (*Figure 3B*, males, Wilcoxon test: W=66; p=0.001 and females, Wilcoxon test: W=36; p=0.0078). Social isolation from adolescence was associated with decreased food intake in the home cage in males ($X^2_{(1, n=22)}$ = 7.2, p = 0.007; *Supplementary*

file 1C-vi), which dampens the overall hypophagic effect of the novel cage (*Figure 3C*, males, Paired t-test: t=2.193; df=10; *p*=0.0531). In contrast, in females, social isolation from adolescence increased the magnitude of the hypophagic response to the novel cage ($X^2_{(1, n=79)}$ = 6.1, p = 0.014; *Supplementary file 1C-iv*) and eliminated variability in the assay (*Figure 3C*, females, Paired t-test: t=6.347; df=7; *p*=0.0004). Body weight was not impacted by the timing of social isolation (data not shown). Notably, sex interacts with the adolescent vs. adult social isolation stress paradigm to modulate both latency and food intake in the home cage (latency: $X^2_{(1, n=47)}$ = 9.7, p = 0.002, *Figure 1—figure supplement 1C-ix*; food intake: $X^2_{(1, n=47)}$ = 6.2, p = 0.013, *Supplementary file 1C-x*) and novel cage (latency: $X^2_{(1, n=47)}$ = 19.4, p < 0.001, *Supplementary file 1C-ix*; food intake: $X^2_{(1, n=47)}$ = 4.2, p = 0.041, *Supplementary file 1C-x*).

We parsed potential contributions of adolescent onset and the length of social isolation. We performed the NSF assay in females that were exposed to prolonged (~6 weeks) social isolation stress as adults (*Figure 3—figure supplement 1A*, red). Latencies were increased in the novel cage (*Figure 3—figure supplement 1B*, red, Wilcoxon test: W=43; *p*=0.00), while hypophagic responses did not reach significance (*Figure 3—figure supplement 1C*, red, Paired t-test: t=2.211; df=8; *p*=0.058). We also examined whether adolescent social isolation for 2 weeks is sufficient to increase reliability of feeding behavior in females. To this end, we performed the NSF assay in 7-week-old females that were singly housed from 5 weeks (*Figure 3—figure supplement 1A*, green). Whereas latencies were increased in the novel cage (*Figure 3—figure supplement 1B*, green, Wilcoxon test: W=85; *p*=0.0012), feeding responses were not consistent (*Figure 3—figure supplement 1C*, green, Wilcoxon test: W=18; *p*=0.2656). In females, the length of social isolation impacted the feeding response, with the longer period promoting hypophagic responses ($X^2_{(1, n=79)}$ = 6.1, p = 0.014; *Supplementary file 1C-iv*), but not latencies ($X^2_{(1, n=79)}$ = 0.1, p = 0.712; *Supplementary file 1C-iii*). In contrast, the timing of social isolation decreased anxiety-like responses ($X^2_{(1, n=79)}$ = 6.3, p = 0.012; *Supplementary file 1D-iii*), but did not impact food intake ($X^2_{(1, n=79)}$ = 0.7, p = 0.421; *Supplementary file 1D-iv*).

In summary, the combination of adolescent onset and prolonged exposure to social isolation is required to obtain consistent and significant hypophagic responses in females, but it has the opposite effect in males (i.e., caloric intake no longer significantly decreased in the novel cage). We defined sex-specific conditions that produce consistent effects of novel stress on feeding behavior in adults – short single housing in males and prolonged single housing from adolescence in females.

## Strategies to increase the relevance of the NSF assay to human physiology

With the ability to study behaviors in both sexes, we next sought to address a major gap between studies of stress-related eating behaviors in humans and rodents (*François et al., 2021*). In the standard NSF assay, an overnight fast is used to motivate mice to eat in the morning, a period when mice typically consume very little food (*Figure 4B*). In contrast, studies in humans deliberately focus on emotional eating in the absence of hunger. We next studied two conditions that can promote food consumption without the physiological stress of fasting – the dark phase of the diurnal cycle and diet palatability.

## Time of day shapes stress responses in a sex-independent manner

Diurnal misalignment has sex-specific effects on systems regulating energy homeostasis in humans that increase susceptibility to obesity (*Qian et al., 2019*). To perform the NSF assay in the period of activity and feeding in mice, we examined behaviors at the onset of the dark phase of the cycle. We used conditions for each sex that produce consistent results in the light cycle – 2-week social isolation in adult males and ~6-week social isolation from adolescence in females (*Figure 4A*, blue and green). As we observed in the light cycle, latencies were increased in the novel cage in both groups (*Figure 4C*, blue and green, Wilcoxon test: W=92; *p*=0.0067 and Wilcoxon test: W=120; *p*=0.0008, respectively). In contrast to the uniform hypophagic response to the novel test in the light phase in males, the same conditions produce mixed results in the dark phase (*Figure 4D*, blue, Wilcoxon test: W=-4.0; *p*=0.9229). In females, longer social isolation produced hypophagic responses (*Figure 4D*, green, Paired t-test: t=2.166; df=15; *p*=0.0468), while shorter social isolation produced hyperphagic responses (*Figure 4D*, red, Wilcoxon test: W=106; *p*=0.0038), as we had observed in the light phase assay. Overall, the time of day had no effect on latency endpoints (*Supplementary file 1E–I, iii*).

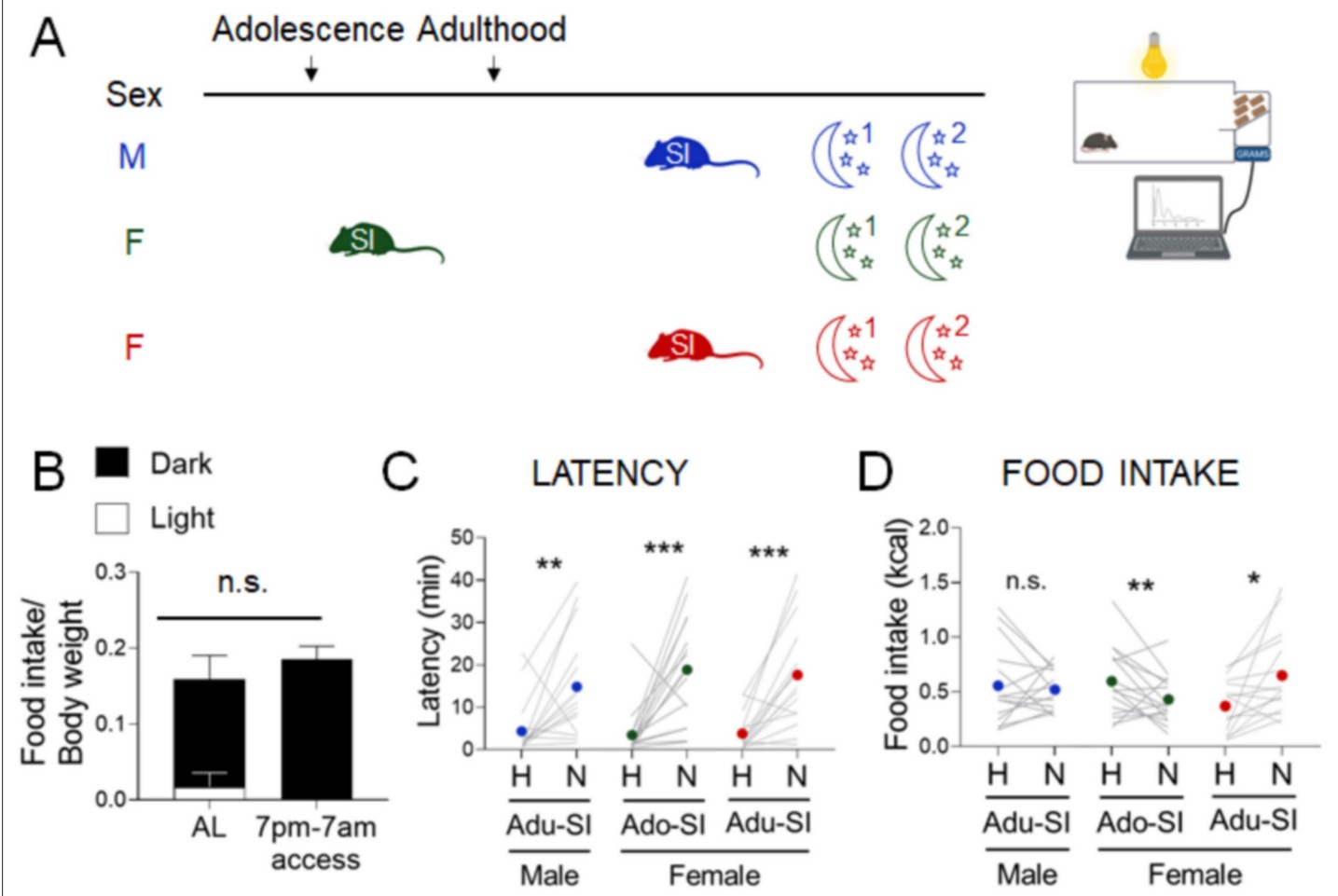

**Figure 4.** Effects of diurnal factors on the novelty-suppressed feeding (NSF) assay in males and females. (**A**) Experimental paradigm for the automated NSF assays performed at the onset of the dark phase in adult mice. (**B**) Food intake per g body weight across the light cycle (white bar) and dark cycle (black bar) of female mice fed ad libitum (AL) or on a 7 pm to 7 am schedule ($n = 7$, paired $t$-test: $t = 1.777$; df = 6; p = 1259). (**C**) Latency to eat in home (H) and novel (N) tests in males socially isolated 2 weeks before the tests (blue, $n = 15$, Wilcoxon test: $W = 92$; p = 0.0067), females socially isolated during adolescence (green, $n = 16$, Wilcoxon test: $W = 120$; p = 0.0008), and females socially isolated 2 weeks before the tests (red, $n = 12$, Wilcoxon test: $W = 126$; p = 0.0003). (**D**) Food intake in home (H) and novel (N) tests in males socially isolated 2 weeks before the tests (blue, $n = 15$, Wilcoxon test: $W = −4.0$; p = 0.9229), females socially isolated during adolescence (green, $n = 16$, paired $t$-test: $t = 2.166$; df = 15; p = 0.0468), and females socially isolated 2 weeks before the test (red, $n = 12$, Wilcoxon test: $W = 106$; p = 0.0038). *p < 0.05, **p < 0.01, ***p < 0.001 between home and novel tests. SI, social isolation; n.s., not significant. See *Figure 4—source data 1*.

The online version of this article includes the following source data and figure supplement(s) for figure 4:

**Source data 1.** Stress assay performed in the dark phase of the light cycle.

**Source data 2.** Parsing the effects of the time of day vs. prandial state.

**Figure supplement 1.** Influences of the estrous cycle when the stress assay is performed in the dark phase of the diurnal cycle.

**Figure supplement 2** Parsing the effects of the time of day vs. prandial state on the effects of novel environment stress on feeding behavior.

Although mice ate less in the dark cycle assay in both home cage ($X^2_{(1, n=88)} = 32.0$, p < 0.001; *Supplementary file 1E-ii*) and novel cage ($X^2_{(1, n=88)} = 10.9$, p = 0.001; *Supplementary file 1E-ii*), responses to stress were shifted toward hyperphagia ($X^2_{(1, n=88)} = 10.1$, p = 0.002; *Supplementary file 1E-iv*).

We asked whether estrous cyclicity influences eating behaviors when the test is performed in the dark phase (undefinedundefined). Since social isolation length does not interact with the estrous cyclicity to influence latency and food intake (*Supplementary file 1C-xiii*, xiv), we pooled both groups to increase statistical power. Latencies to eat were not affected by estrous stage in the home cage (*Figure 4—figure supplement 1A*, Kruskal-Wallis test: $H_{(2)} = 0.5602$; p=0.7557), but were significantly

reduced in proestrus in the novel cage (*Figure 4—figure supplement 1B*, One-way ANOVA: $F_{(2, 25)}$=1.197; $p$=0.0053). In contrast, food intake was reduced in estrus compared to diestrus in the home cage (*Figure 4—figure supplement 1C*, Kruskal-Wallis test: $H_{(2)}$=7.649; $p$=0.0218), but there was no effect of cyclicity in the novel cage (*Figure 4—figure supplement 1D*, Kruskal-Wallis test: $H_{(2)}$=0.4612; $p$=0.7941).

Because mice consume ~90% of their caloric intake in the dark phase, the test performed in the morning after an overnight fast imposes a state of negative energy balance, while daily intake is not reduced in mice with restricted access to food in the dark phase (*Figure 4B*).

To parse the effects of diurnal influences vs. prandial state, we matched the 90% caloric restriction achieved by an overnight fast by allowing mice to consume an equivalent number of calories (~2 kcal) at the start of the dark cycle on the previous day (*Figure 4—figure supplement 2A,B*). Latency outcomes were not impacted by the fast (*Figure 4C*, Paired t-test: t=2.932, df=7; $p$=0.02, and *Supplementary file 1E–I, iii*). While food intake was consistently and significantly decreased in the novel cage during the light phase test (*Figure 3C*, green, Paired t-test: t=6.347; df=7; $p$=0.0004), this effect was lost in the dark cycle test (*Figure 4—figure supplement 2D*, Paired t-test: t=2.104, df=7; $p$=0.0734). Therefore, diurnal influences, and not prandial state, are the primary determinants of whether stress increases or decreases intake.

## Chronic HFD consumption produces hyperphagic stress responses in both sexes

Next, we investigated whether elevated body weight also increases the likelihood of hyperphagic responses, as has been observed in humans (*Laitinen et al., 2002*; *Coulthard et al., 2021*; *Lemmens et al., 2011 Figure 5A*). Mice exposed to HFD for 10–12 weeks increased body weight (*Figure 5B*, males, Paired t-test: t=11.25; df=12; $p$<0.0001 and females, Paired t-test: t=13.53; df=13; $p$<0.0001) but were not frankly diabetic (*Figure 5C*). Males increased latency in the novel cage (*Figure 5D*, Paired t-test: t=4.751; df=12; $p$=0.0005), but females did not (*Figure 5D*, Wilcoxon test: W=5; $p$=0.9032). Under these conditions, both males and females increased their caloric intake in the novel cage (*Figure 5E*, males, Wilcoxon test: W=89; $p$=0.0005 and females, Wilcoxon test: W=105; $p$=0.0001). Body weight gain was not correlated with the change in food intake between the home and novel cage (*Figure 5F*, Linear regression: $R^2$=0.01017; $F_{(1, 25)}$=0.2568; $p$=0.6167). In summary, stress-induced hyperphagia in the dark phase assay was exacerbated by chronic exposure to HFD in both sexes ($X^2_{(1, n=58)}$ = 11.8, p = 0.001; *Supplementary file 1F-iv*).

We next parsed the effects of chronic exposure to HFD vs. access to a highly palatable diet during the test (*Figure 5—figure supplement 1A*). Mice were acclimated to HFD for 30 min per night for 5 days, which did not affect body weight (*Figure 5—figure supplement 1B*, males, Paired t-test: t=1.266; df=10; $p$=0.2343 and females, Paired t-test: t=1.389; df=15; $p$=0.1865). Males increased latency in the novel cage (*Figure 5—figure supplement 1C*, Wilcoxon test: W=46; $p$=0.04), but females did not (*Figure 5—figure supplement 1C*, Paired t-test: t=0.6366; df=14; $p$=0.5347), as observed in chronic HFD exposure. As anticipated, lean mice had tenfold lower latencies ($X^2_{(1, n=56)}$ = 28.5, p < 0.001; *Supplementary file 1G-iii*) and consumed >10 times more calories of the palatable HFD than the chow in the home cage ($X^2_{(1, n=56)}$ = 398.2, p < 0.001; *Supplementary file 1G-ii*). They also ate more of the HFD than the overweight group ($X^2_{(1, n=51)}$ = 29.8, p < 0.001; *Supplementary file 1G-x*). Males decreased food intake in response to the novel cage stress (*Figure 5—figure supplement 1D*, Paired t-test: t=3.956; df=10; $p$=0.0027), while females did not (*Figure 5—figure supplement 1D*, Paired t-test: t=0.5385; df=14; $p$=0.5987). In summary, chronic exposure to HFD, and not the palatability of the test diet, is the primary driver of the hyperphagic response to stress observed in both males and females.

## Discussion

Rodent models typically use stress-induced hypophagia as a surrogate for anxiety- or depression-like behaviors in males, whereas studies in humans commonly focus on emotional overeating as a risk factor for obesity, with a bias toward women. Our studies narrow this gap. We defined sex-specific variations of the NSF assay that permit investigations of consistent stress-related feeding behaviors in males and females. In addition, performing the assays in the active phase of the light cycle and in

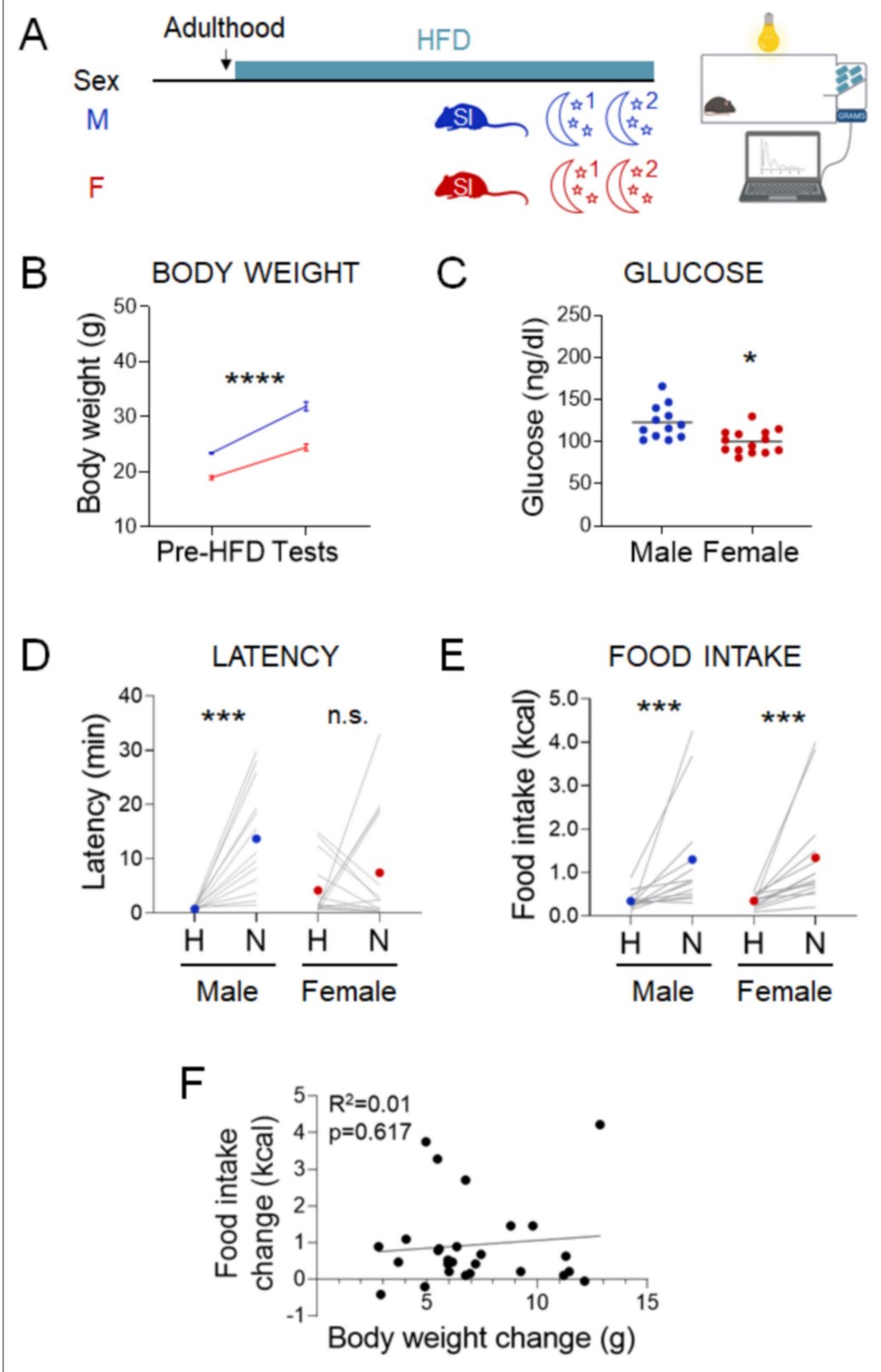

**Figure 5.** Effects of chronic exposure to high-fat diet (HFD) on the novelty-suppressed feeding (NSF) assay in males and females. (**A**) Experimental paradigm for the automated NSF assay performed at the onset of the dark phase in adult chronically exposed to HFD mice socially isolated 2 weeks before the tests. (**B**) Body weights before exposure to HFD and at the time of the tests, in males (blue, *n* = 13, paired *t*-test: *t* = 11.25; df = 12; p <

*Figure 5 continued on next page*

*Figure 5 continued*

0.0001) and females (red, $n$ = 14, paired *t*-test: $t$ = 13.53; df = 13; p < 0.0001). (**C**) Blood glucose levels in males (blue, $n$ = 13) and females (red, $n$ = 14, unpaired *t*-test: $t$ = 3.460; df = 24; p = 0.002). (**D**) Latency to eat in home (H) and novel (N) tests in adult males (blue, $n$ = 13, paired *t*-test: $t$ = 4.751; df = 12; p = 0.0005) and females (red, $n$ = 14, Wilcoxon test: $W$ = 5; p = 0.9032). (**E**) Food intake in home (H) and novel (N) tests in adult males (blue, $n$ = 13, Wilcoxon test: $W$ = 89; p = 0.0005) and in adult females (red, $n$ = 14, Wilcoxon test: $W$ = 105; p = 0.0001). (**F**) Correlation between change in food intake in N vs. H and change in body weight before and after chronic exposure to HFD. *p < 0.05, ***p < 0.001, ****p < 0.0001. SI, social isolation; n.s., not significant. See *Figure 5— source data 1*.

The online version of this article includes the following source data and figure supplement(s) for figure 5:

**Source data 1.** Stress assay performed in the dark phase of the light cycle in overweight mice chronically exposed to high-fat diet (HFD).

**Source data 2.** Stress assay performed in the dark phase of the light cycle in lean mice acutely exposed to high-fat diet (HFD).

**Figure supplement 1.** Effect of novel envrionment stress assessed in the dark phase of the light cycle in lean mice acutely exposed to high-fat diet (HFD).

the context of chronic HFD consumption shifts responses toward hyperphagia, which more faithfully recapitulates the bias toward overeating in people (*Oliver and Wardle, 1999*; *Phillipou et al., 2020*; *Kandiah et al., 2006*). Finally, discordance between effects of some experimental parameters on anxiety-like and feeding behaviors could indicate that the underlying circuits are distinct.

## Assessing stress-induced behaviors: technical considerations

By focusing exclusively on latency as a readout of anxiety-like behavior in the NSF, investigators miss the opportunity to evaluate the effect of stress on feeding behavior. We incorporated several features into the basic NSF paradigm to permit reliable and meaningful assessments of caloric intake. We trained mice to eat under the test conditions for several days until baseline levels of intake stabilized and excluded those that failed to train (*Schalla et al., 2020*). We calculated intake from the first bite and not the start of the test. This is critical, as the high novel cage latencies (≥15 min) exhibited by most of the female groups and the overweight males would confound interpretation of feeding measurements that started at the onset of the test. We evaluated intake for 30 min, as recommended (*Schalla et al., 2020*; *Ellacott et al., 2010*). Finally, analyzing both baseline and stress-induced conditions makes it possible to parse the effects of acute stress from those imposed by stressors built into the paradigm. This is important because we observed the most variability in chow-based studies females in the home cage. Some factors, such as adolescent social isolation, have opposite effects on behavior in the home and novel cages. Uncovering experimental conditions that influence behavior across the estrous cycle in the home cage helped to define conditions that produce consistent behavioral outcomes in females.

## Circuits mediating stress-induced anxiety-like and feeding behaviors are likely distinct

The novel cage consistently induced higher latencies, except for females acutely or chronically exposed to HFD. In contrast, feeding responses to stress were variable between groups. Moreover, distinct sets of factors influenced latency and feeding outcomes. The manual test and the timing of the onset of social isolation affected only latency outcomes, while the length of social isolation, time of day, and chronic HFD exposure only impacted feeding outcomes. The only feature of the paradigm that influenced both latency and food intake was acute exposure to the palatable diet. Together, these observations support the idea that the circuits regulating stress-induced anxiety-like and feeding behaviors are likely distinct.

## Influences of sex and estrous cyclicity on stress-related feeding behaviors

Sex differences in feeding behavior are well documented (*Asarian and Geary, 2013*). Daily food intake fluctuates across the estrous cycle, with reductions during proestrus, when estrogen is at its peak, in rodents and humans (*Tarttelin and Gorski, 1971*; *Petersen, 1976*; *Buffenstein et al., 1995*). In contrast, fasting-induced food intake was significantly higher in proestrus than in diestrus, consistent with studies involving estrogen replacement in ovariectomy (*Shakya and Briski, 2017*). Low home cage intake after an overnight fast in diestrus likely reflects increased stress, because it is restored by automated measurements. When we eliminated the fast by performing the test in the dark cycle, the anorexigenic effects of estrogen were retrieved, as reported by others (*Asarian and Geary, 2013*). Performing automated measurements eliminated effects of the estrous cycle on baseline intake but did not reduce the heterogeneity of feeding responses to stress. Conversely, we observed consistent hyperphagic responses in the dark cycle in the face of variability in home cage intake across the estrous cycle. Therefore, our studies debunk the assumption that estrous cyclicity drives variability in stress responses in females that precludes their use in neurobehavioral studies.

## Sex-dependent effects of social isolation on stress-related feeding behaviors

Social isolation in adulthood produces opposite effects in males and female feeding behavior in the NSF, with males more likely to exhibit hypophagic behavior (*Oliver et al., 2020*). Moreover, exposure to social isolation in the postweaning period has sex-specific effects on neurobehavioral outcomes, with increased sensitivity in males (*Weiss et al., 2004*; *Fone and Porkess, 2008*). Here, we examined the impact of social isolation in adolescence, because it is a sensitive period for programming stress responses (*Wright et al., 1991*) and the peak onset of eating disorders (*Jacobi et al., 2004*; *Bulik, 2002*). Longer exposure to social isolation influenced feeding behavior in a sex-dependent manner, promoting hypophagia in females. Exploring whether the impacts of adolescent social isolation on stress-induced feeding behavior are permanently programmed or can be reversed is an important area for future research.

## Increasing physiological relevance to humans

Assessing behaviors in the dark cycle is more physiologically relevant for feeding and minimizes stresses associated with the external environment and fasting, which impose sex-specific effects in mice and humans (described above). However, reported effects of diurnal factors on anxiety-like behavior are inconsistent, and are sex and assay dependent (*Richetto et al., 2019*; *Nakano et al., 2016*). Here, we found that performing the NSF assay at the start of the dark cycle promoted hyperphagic responses in both sexes. This could stem from a lower appetitive drive in the home test compared to an overnight fast. Diurnal influences, and not prandial state, are also more important influences on food intake in rats (*Schalla et al., 2020*). The high degree of variability in food intake responses in the dark cycle in lean females and males is similar to what is seen in humans (*Wardle et al., 2011*).

In men and women, emotional eating is most commonly observed in the context of 'comfort' foods (*Wallis and Hetherington, 2009*; *Rutters et al., 2009*; *Kandiah et al., 2006*; *Oliver et al., 2000*; *Zellner et al., 2006*) and is associated with elevated BMI (*Laitinen et al., 2002*; *Lemmens et al., 2011*; *Cohen et al., 2002*). Consistent with these observations, we found that chronic exposure to HFD promoted hyperphagic responses to novel environment stress in both males and females. Stress-induced anxiety-like and feeding behaviors in females exposed to chronic HFD were discordant, consistent with observations in humans (*Wardle et al., 2011*). These effects are likely the consequence of the chronic exposure to HFD in contrast to the diet palatability, as performing the assay with HFD in lean mice produced opposite effects in both sexes. Although we did not observe correlations between body weight gain and feeding responses, identifying neuroendocrine or neuronal biomarkers that predict stress-induced hypophagia vs. hyperphagia is an important topic for future research.

## Summary

We identified factors that have predictable effects on stress-induced feeding responses in both sexes (*Figure 6*). Performing the assay under conditions that are more physiologically relevant for humans,

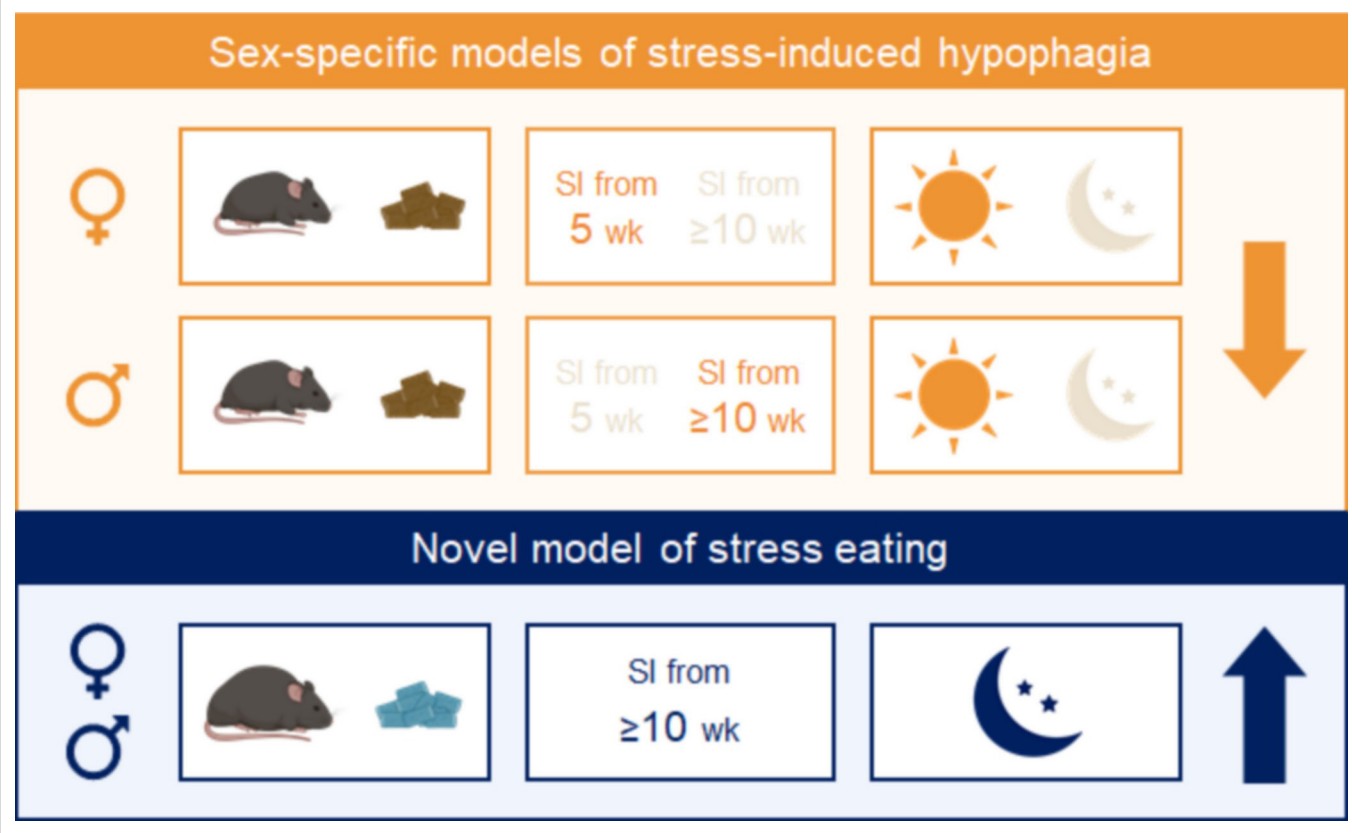

**Figure 6.** Summary. Combining experimental variables that influence stress-induced food intake in the same direction yields consistent and reproducible effects in males and females. Sex-specific models of stress-induced hypophagia (top panel). The lean state and performing the assay in the light phase promote hypophagic responses. A short period of social isolation in adult (≥10 weeks) males elicits hypophagic responses, but a prolonged (~6 weeks) period of social isolation starting in adolescence (5 weeks) is needed to produce the same effect in females. Sex-independent model of stress-induced hyperphagia (bottom panel). Chronic exposure to high fat diet and performing the assay in the dark phase promote hyperphagic responses in both sexes.

during the active phase and in conjunction with chronic HFD exposure, promotes hyperphagic responses. On the other hand, conducting the test in the inactive phase in lean mice, conditions used in most behavioral assays, promotes hypophagic responses. The duration of social isolation exerted sex-dependent effects. The use of other stress paradigms and other diets will be critical to draw more generalized conclusions. These studies can serve as a framework to develop sex-specific variations of paradigms to model subclinical and disordered eating behaviors in humans.

## Materials and methods
### Animals
C57BL/6J wild-type male and female mice (WT, Jax strain #000664, RRID:IMSR JAX:000664) were maintained on a 12 hr/12 hr light/dark cycle, with ad libitum access to food and water, unless stated otherwise. Tests were performed in adult (~12–30 weeks) or adolescent (7 weeks) mice, fed either a standard chow diet (PicoLab Rodent Diet) or a HFD (D12492, 60% fat, 20% sugar, Research Diets). All procedures were performed within the guidelines of the Institutional Animal Care and Use Committee (IACUC) at the Columbia University Health Science Division.

### NSF assay
The impact of novel stress was evaluated by comparing behaviors in the familiar environment of the home cage (home test, H) vs. a new cage without bedding, under bright lighting, and lined with white paper (novel test, N). The outcome measures were latency to eat and the amount consumed within

30 min from the first bite. Mice did not have access to water during the tests. Body weight and daily food intake were monitored throughout the study. The bedding was not changed throughout the length of the experiment.

## Manual measurements

Mice were acclimated to the test conditions for 30 min every morning for five consecutive days. Their home cages were moved to a bench, where they were trained to eat chow from the floor in the open cage in the presence of the experimentalist. Mice that did not train to eat from the floor were excluded from further analyses. Both tests were performed following an overnight fast (~16 h). Mice were allowed to recover for 3–4 days with additional training sessions between the two tests.

## Automated measurements

To minimize stressors associated with manual measurements, such as cage transport, lid opening, and human interaction, we adapted the assay to an automated food intake monitoring system (BioDAQ, Research Diets) (*Schalla et al., 2020*). Mice were acclimated to the BioDAQ cages and food hoppers for 5–7 days. Mice that did not train to eat or drink from the BioDAQ hoppers were excluded from further analyses. Daily food intake, latency to eat and 30 min caloric intake (after the first bite) were assessed using the BioDAQ Dataviewer software. Home and novel tests were counterbalanced and performed 1–3 days apart.

## Variables in experimental paradigms

### Social isolation stress

Mice were singly housed in adulthood (>10 weeks of age) or during mid-adolescence (5 weeks of age). We controlled for different lengths of social isolation periods by including groups that were singly housed for 2 or ~6 weeks.

## Time of day

To study behaviors at the start of the dark phase, precise timing of access to food was provided by programming automated gates to open at the start of the dark cycle (7 pm). After acclimating to the BioDAQ system for 3–4 days, mice were trained to a schedule of restricted access to food from 7 pm to 7 am for 5–7 days. In one cohort, we controlled for the severity of the fast associated with the light cycle test, by performing the assay in the dark phase in adult females that were exposed to a 90% caloric restriction by limiting access to the food from 7 pm to 8 pm on the previous day.

## Diet palatability

Mice were given access to HFD at 7 pm for 30 min during 5 days prior to the tests, as well as during the home and novel tests.

## Chronic exposure to HFD

Mice were given access to both chow and HFD for 8 weeks in males and 10 weeks in females before the 2-week training and acclimation period to the BioDAQ. Body weight was monitored weekly. Once in the BioDAQ, mice had access to HFD only. Mice that did not eat the HFD during the first hour of the dark cycle during acclimation were excluded from further analysis.

## Hormone measurements

Mice were euthanized by an overdose of isoflurane followed by decapitation, at the end of either home or novel tests (30–45 min after onset of eating). Trunk blood was collected in Microvette gel tubes (Nalgene, 5000-1012) and centrifuged at 10,000 × $g$ for 15 min. Serum was stored at −80°C until further analysis. Serum levels of luteinizing hormone (LH), follicle-stimulating hormone (FSH), and β-estradiol were measured by the Ligand Assay and Analysis Core at the University of Virginia Center for Research in Reproduction, using the Millipore Pituitary Panel Multiplex kit for LH/FSH, and the Calbiotech Mouse/Rat Estradiol ELISA kits.

## Vaginal cytology and estrous phase determination

Vaginal swabs were collected at the end of each test (in both home and novel conditions), and estrous cycle phases were classified by cytology (*McLean et al., 2012*). Criteria for classification were: *Diestrus*: a mix of cornified epithelial cells and leukocytes (early, or metestrus), or leukocytes only (late); *Proestrus*: a mix of leukocytes and nucleated epithelial cells (early), or a majority of nucleated epithelial cells (late); *Estrus*: a mix of nucleated epithelial cells and cornified epithelial cells (early) or a majority of cornified epithelial cells (late) (*Figure 1—figure supplement 1A*). High LH levels were detected in 4 of 25 proestrus females (*Figure 1—figure supplement 1B*). FSH levels were significantly higher in proestrus females (*Figure 1—figure supplement 1C*). Estradiol levels were similar across all phases of the estrous cycle (*Figure 1—figure supplement 1D*).

## Statistical analyses

Intragroup statistical tests were performed with GraphPad Prism software. Grubb's test was used to detect outliers in each experimental group. Outliers were excluded from all analyses. The distribution of values for latency, food intake, body weight, and glucose were assessed with the Shapiro–Wilk normality test. Differences in behaviors between home and novel environments were subsequently analyzed with either a paired Student's *t*-test or Wilcoxon matched pairs test, as appropriate for the distribution. Differences between phases of the estrous cycle were analyzed with one-way analysis of variance and Tukey's multiple comparison post-test, or Kruskal–Wallis and Dunn's multiple comparison post-tests, according to the distribution. Pearson's correlation was used to measure the relationship between change in food intake from the home to the novel cage and gain of body weight during chronic exposure to HFD. A 95% confidence interval was used to determine significance, which was reported on graphs using $*p < 0.05$, $**p < 0.01$, $***p < 0.001$, and $****p < 0.0001$.

We built regression models to analyze variables across groups using SPSS 28.0 (RRID:SCR_002865). In each of the two cage types (i.e., home cage and novel cage), we used a generalized line model (GLM) with identity link function to examine the association between potential risk factors (such as sex, social isolation, high fact diet exposure, and estrus cycle) and the two primary outcomes (i.e., latency and food intake). In addition to constant term and the risk factors, the analysis model also included body weight, age, and testing order to adjust for potential confounding. We conducted the analysis in the full sample as well as several specific subgroups of interest using the same analytic approach. We then compared the strength of association between the risk factors and outcomes by cage type using GLM. Generalized estimating equation methodology with exchangeable working correlation matrix and robust variance estimator was employed to account for the within-mouse correlation due to repeated measure of outcomes from the same mouse. For such analysis, the statistical model included the constant term, risk factor, cage type (novel vs. home), the cage type-by-risk factor interaction, and the potential confounding factors (i.e., body weight, age, and testing order). Findings with p values <0.05 were declared as statistically significant.

## Acknowledgements

We thank the technical support team at ResearchDiets for guidance in setting up the BioDAQ system. Microscopy was performed in the Columbia Diabetes Research Center Core P30 DK063608. Diagrams were generated with BioRender. Reproductive hormone assays were performed by the University of Virginia Center for Research in Reproduction, Ligand Assay and Analysis Core, supported by the Eunice Kennedy Shriver NICHD Grant R24 HD102061.

## Additional information

### Funding

| Funder | Grant reference number | Author |
| --- | --- | --- |
| National Institute of Mental Health | 1R01 MH113353 89038 | Lori Zeltser |
| Klarman Family Foundation | | Lori Zeltser |

| Funder | Grant reference number | Author |
|---|---|---|
| Russell Berrie Foundation | | Lori Zeltser |
| National Institute of Child Health and Human Development | R24 HD102061 | Lori Zeltser |

The funders had no role in study design, data collection, and interpretation, or the decision to submit the work for publication.

## Author contributions

Marie Francois, Formal analysis, Investigation, Methodology, Writing – original draft; Isabella Canal Delgado, Investigation, Visualization; Nikolay Shargorodsky, Investigation; Cheng-Shiun Leu, Formal analysis; Lori Zeltser, Conceptualization, Funding acquisition, Project administration, Supervision, Writing - review and editing

## Author ORCIDs

Marie Francois  http://orcid.org/0000-0002-0418-6282
Nikolay Shargorodsky  http://orcid.org/0000-0002-9705-2645
Lori Zeltser  http://orcid.org/0000-0002-4330-6240

## Ethics

This study was performed in strict accordance with the recommendations in the Guide for the Care and Use of Laboratory Animals of the National Institutes of Health. All procedures were performed within the guidelines of the Institutional Animal Care and Use Committee (IACUC) at the Columbia University Health Science Division (protocols #AC-AABN3551 and AC-AABN3553).

## Decision letter and Author response

Decision letter https://doi.org/10.7554/eLife.70271.sa1
Author response https://doi.org/10.7554/eLife.70271.sa2

# Additional files

## Supplementary files

• Supplementary file 1. Results of multivariable statistical analyses across groups. (A) Summary of experimental groups. Overview of the number of cohorts and animals included in the analysis of each experimental type, as well as the number of animals excluded because they failed to train or were outliers. (B) Test type. Strength of association between two main outcomes, latency (i) and food intake (ii), and type of the assay (manual vs. automated) in the morning assay in females on chow, socially isolated in adulthood. Strength of association between the two outcomes, latency (iii) and food intake (iv), by test type (novel vs. home), and type of the assay, in the morning assay in females on chow, socially isolated in adulthood. The models were adjusted for body weight, age, and test order (home or novel test performed first). (C) Social isolation length. Strength of association between two main outcomes, latency (i) and food intake (ii), and length of social isolation (2 vs. 6–7 weeks) in both morning and dark phase assays in females on chow. Strength of association between the two outcomes, latency (iii) and food intake (iv), by test type (novel vs. home), and length of social isolation, in both morning and dark phase assays in females on chow. Strength of association between two main outcomes, latency (v) and food intake (vi), and length of social isolation (2 vs. 6–7 weeks) in the morning msuppleemanual assay in males on chow. Strength of association between the two outcomes, latency (vii) and food intake (viii), by test type (novel vs. home), and length of social isolation, in the morning manual assay in males on chow. Strength of association between two main outcomes, latency (ix) and food intake (x), sex, and length of social isolation (2 vs. 6–7 weeks) in the morning manual assay in adult males and females on chow socially isolated at 5 weeks for 7 weeks or during adulthood for 2 weeks. Strength of association between the two outcomes, latency (xi) and food intake (xii), by test type (novel vs. home), sex, and length of social isolation, in the morning manual assay in adult males and females on chow socially isolated at 5 weeks for 7 weeks or during adulthood for 2 weeks. Strength of association between two main outcomes, latency (xiii) and food intake (xiv), estrous cycle, and length of social isolation (2 vs. 6–7 weeks) in adult females on chow socially isolated at 5 weeks for 7 weeks or during adulthood for 2 weeks. The models were adjusted for body weight, age, and test order (home or novel test performed first). (D) Social

isolation timing. Strength of association between two main outcomes, latency (i) and food intake (ii), and timing of social isolation (5-week-old vs. adulthood) in both morning and dark phase assays in females on chow. Strength of association between the two outcomes, latency (iii) and food intake (iv), by test type (novel vs. home), and timing of social isolation, in both morning and dark phase assays in females on chow. The models were adjusted for body weight, age, and test order (home or novel test performed first). (E) Time of day. Strength of association between two main outcomes, latency (i) and food intake (ii), and time of day (10 am vs. 7 pm) in adult females on chow (mice socially isolated at 5 weeks or for 7 weeks in the morning assay excluded). Strength of association between the two outcomes, latency (iii) and food intake (iv), by test type (novel vs. home), and time of day (10 am vs. 7 pm) in adult females on chow (mice socially isolated at 5 weeks or for 7 weeks in the morning assay excluded). The models were adjusted for body weight, age, and test order (home or novel test performed first). (F) Chow vs. chronic HFD. Strength of association between two main outcomes, latency (i) and food intake (ii), and type of diet (chow vs. chronic HFD) in the dark phase assay in adult males and females socially isolated in adulthood. Strength of association between the two outcomes, latency (iii) and food intake (iv), by test type (novel vs. home), and type of diet (chow vs. chronic HFD) in the dark phase assay in adult males and females socially isolated in adulthood. The models were adjusted for body weight, age, and test order (home or novel test performed first). (G) Acute HFD. Strength of association between two main outcomes, latency (i) and food intake (ii), and type of diet (chow vs. acute HFD) in the dark phase assay in adult males and females socially isolated in adulthood. Strength of association between the two outcomes, latency (iii) and food intake (iv), by test type (novel vs. home), and type of diet (chow vs. acute HFD) in the dark phase assay in adult males and females socially isolated in adulthood. Strength of association between two main outcomes, latency (v) and food intake (vi), sex, and type of diet (chow vs. acute HFD) in the dark phase assay in adult males and females socially isolated in adulthood. Strength of association between the two outcomes, latency (vii) and food intake (viii), by test type (novel vs. home), sex, and type of diet (chow vs. acute HFD) in the dark phase assay in adult males and females socially isolated in adulthood. Strength of association between two main outcomes, latency (ix) and food intake (x), and length of HFD (acute vs. chronic) in the dark phase assay in adult males and females socially isolated in adulthood. Strength of association between the two outcomes, latency (xi) and food intake (xii), by test type (novel vs. home), and length of HFD in the dark phase assay in adult males and females socially isolated in adulthood. Strength of association between two main outcomes, latency (xiii) and food intake (xiv), sex, and length of HFD in the dark phase assay in adult males and females socially isolated in adulthood. Strength of association between the two outcomes, latency (xv) and food intake (xvi), by test type (novel vs. home), sex, and length of HFD in the dark phase assay in adult males and females socially isolated in adulthood. The models were adjusted for body weight, age, and test order (home or novel test performed first). (H) Sex. Strength of association between two main outcomes, latency (i) and food intake (ii), and sex in mice on chow (females in the automated morning assay, adolescent females and adult females socially isolated for 6–7 weeks excluded). Strength of association between the two outcomes, latency (iii) and food intake (iv), and sex in mice on chow (females in the automated morning assay, adolescent females and adult females socially isolated for 6–7 weeks excluded). Strength of association between two main outcomes, latency (v) and food intake (vi), and sex in all mice on HFD. Strength of association between the two outcomes, latency (iii) and food intake (iv), and sex in all mice on HFD. The models were adjusted for body weight, age, and test order (home or novel test performed first). (I) Estrous cycle. Strength of association between two main outcomes, latency (i) and food intake (ii), and phase of the estrous cycle (Diestrus, Proestrus, and Estrus) in all females on chow and socially isolated in adulthood. Strength of association between two main outcomes, latency (iii) and food intake (iv), and phase of the estrous cycle (Diestrus, Proestrus, and Estrus) in all females on acute or chronic HFD. The models were adjusted for body weight, age, and test order (home or novel test performed first).

• Transparent reporting form

### Data availability

All data generated or analysed during this study are included in the manuscript and supporting file; Source Data files have been provided for Figures 1-5.

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
