## [Editor Report]

All of the reviewers have found your response to the previous critiques thoughtful and thorough. Your studies will serve as a valuable resource to the field and will be widely cited.

---

## [Decision Letter]

**Decision letter after peer review:**

Thank you for submitting your article "Assessing effects of stress on feeding behaviors in laboratory mice" for consideration by *eLife*. Your article has been reviewed by 4 peer reviewers, one of whom is a member of our Board of Reviewing Editors, and the evaluation has been overseen by Ma-Li Wong as the Senior Editor. The following individual involved in review of your submission has agreed to reveal their identity: Joseph Bass (Reviewer #2).

Essential revisions:

1) The authors should assess the question of whether palatability might impact latency of anxiety behavior and/or energy intake.

2) The authors need to reassess their statistical analyses. It is advisable to recruit a biostatistical collaborator to potentially build a more informative multivariable model. In the absence of this more robust analysis, the strength of some conclusions are questionable and the authors should focus on the within-experiment comparisons.

3) Another and related concern is the sample sizes. More mice are needed.

4) Another concern is the focus only on % of study subjects that respond in one way or another as opposed to the magnitude of the changes. This needs to be addressed.

5) Comparisons of variance in the feeding response are needed to support the conclusions regarding the effect of a given variable on the likelihood of changing food intake in response to stress.

6) The inclusion of other anxiety-like tests is required as fasting modulates anxiolytic effects and this effect increases with the fasting time length (Changhong L et al., 2019; Heinz DE et al., 2021) -- This confounding needs to be addressed.

7) The use of another stress paradigm would be required to validate the major points of the work.

*Reviewer #1:*

Francois and colleagues describe results from a series of studies assessing the effects of stress on feeding behavior in mice. Notably, the authors provided detailed assessment of various parameters while taking into account sex, age, circadian and estrous cycles. The studies are very well described and provide data that will be of wide interest.

The focus on anxiety and eating is well justified. The potential link to eating disorders in humans is briefly mentioned. It would be useful if the current results in context of sex differences and eating disorders would be welcome.

*Reviewer #2:*

This work addresses a range of variables and contradictory findings in analyses of the anxiety-feeding interface by rigorously evaluating several parameters influencing feeding behaviors. While clinical evidence shows susceptibility to either hyperphagia or hypophagia that is predictable and sexually-dimorphic at an individual level, population conclusions have been more difficult to predict. Specifically, factors such as sex and BMI have been implicated in human over- or under-eating under stress; however, a clear picture has not emerged in experimental animal studies due to lack of evaluation and/or standardization of these parameters. Francois et al., systematically evaluate how developmental (and 24h) timing of social isolation impacts feeding in a sex-dependent manner, identify daily timing of novelty-induced feeding behaviors. Their work also probes how diet-induced obesity impacts feeding responses under stress and provide recommendations for creating predictable responses in animals that allow for best practices in studies of stress-induced feeding. Further, they identify that proper design of these experiments can lead to much greater predictability of feeding responses, indicating that the benchmarks herein will enlighten future neurocircuit-level analyses of stress-related appetitive behaviors.

While the behavioral characterizations were robustly conducted and provide important new insights to calibrate the entire and broad field of obesity research, a few considerations would enhance the final form of this manuscript. In particular, the question of whether palatability might impact latency of anxiety behavior and/or energy intake, and whether such factors may change according to nutrient prehistory may be helpful (e.g., are DIO mice driven more via rival hedonic vs homeostatic responses). Is there any way to parse out more within the present work whether the latency phenomenon vs energy intake aspect might reflect differences in "homeostatic drive": for instance, are leptin(or active ghrelin) levels (or leptin-supressed refeeding responses) different amongst males/females, preadolescent/vs. post-pubertal, normal/vs. high-fat fed animals? The point would be to define with endocrine or biochemical quantification the observed distinction between latency and food intake endpoints. Equally exciting would be the observation that ACTH/cortisol levels might vary in a way that could provide a similar entry point for future endocrine analyses (or at least for these points one might review relevant existing literature to provide context from the starting point here).

– Addition of more behavioral assays or discussion of such an approach may extend context for the NSF paradigm.

– Question: With regard to social isolation and 24h timing in males vs females, is it possible that females are more synchronized when group-housed--would difference in estrus days explain variance within female experiments?

– As presented the work analyzed animals under light:dark conditions, so in formal terms the observations describe "diurnal" aspects of behavior rather than "circadian"-the latter requiring analyses under constant conditions such 24h dark:dark (a semantic point that may be a consideration to those in the field in the future).

*Reviewer #3:*

In this report, Marie Francois and colleagues set out to uncover aspects of experimental paradigms that promote hyperphagic vs hypophagic responses to psychological stress in mice. The comprehensive experiments were built upon the novelty-suppressed feeding (NSF) test, that is commonly used as an index of anxiety-like behavior. The authors make the salient point that collecting caloric intake data in this paradigm could expand the information revealed by the assay and potentially provide a better understanding of how, and under what experimental conditions, acute stress increases vs decreases caloric intake in mice. They performed 13 variations of the NSF test, to determine the influence of sex, estrous cycle, time of day, age and duration of social isolation, and diet on anxiety like and feeding behavior.

The data strongly support that latency to eat in the novel environment, an index of anxiety, was robustly increased across experimental conditions and in both sexes. By contrast, effects on feeding behavior were more variable. Most convincingly, in Figure 4, the authors demonstrate that females socially isolated for ~6 weeks beginning in adolescence exhibit novelty-suppressed caloric intake whereas females socially isolated for 2 weeks beginning in adulthood exhibit novelty-exaggerated caloric intake, when the test is performed during the normal feeding period at the onset of dark. Although the relative contributions of length of isolation and age of isolation onset remain entangled, these findings point towards two key variables that may influence the well-characterized divergent response to stress-eating in women setting the stage for mechanistic studies. Another factor that may influence the direction of the feeding response to stress is obesity, since DIO male and female mice exhibited a hyperphagic response to the novel environment. This conclusion could be strengthened by a direct comparison between lean and obese mice in the same experiment.

Strengths:

– The important research question and creative approach.

– Careful attention to rigor in experimental design including counterbalancing the home cage and novel cage tests, training mice to eat under the test conditions, measuring caloric intake for 30 minutes from the first bite to avoid confounding latency and caloric intake, attention to estrus cycle as a potential variable, and attention to degree of caloric restriction as a potential variable.

– Use of the automated BIODAQ feeding system represents an important innovation allowing for feeding to be assessed under more ethologically-relevant conditions.

– Clear and easy to digest figures and methods.

Weaknesses:

The primary concern is regarding the statistical design and analysis.

– Essentially, the authors have created a powerful, large multifactorial experimental design but there is a missed opportunity to extract robust information since analyses are not integrated in a single model. Rather, there are mostly a series of pairwise comparisons, some of which span experiments.

– By my understanding, the data were collected across ~10 individual experiments represented by the various figures, and presumably within each experiment the groups were assayed simultaneously (e.g., in Figure 4 females isolated in adolescence and females isolated in adulthood were tested in the NSF test at the same time). However, many of the statistical comparisons shown in Supplemental Table 1 and resulting conclusions occur across experiments. There is less confidence in these conclusions/ findings since behavioral responses to stress are extremely labile and easily influenced by stochastic environmental conditions, which are not controlled when comparing across experiments.

– Comparisons of *variance* in the feeding response are critical to several conclusions regarding the effect of a given variable on the 'likelihood of hyperphagic vs hypophagic responses to stress'. However these comparisons are generally only narrative, rather than employing statistical tests to compare variance (or likelihood) between groups, and they also rely on binning the change in caloric intake to categorical variables rather than presenting the data as a continuous variable.

Additional physiological endpoints could begin to inform mechanism. For example, how the various timing(s) of social isolation affect body weight and whether weight is a significant overall predictor of the feeding response to stress. Likewise, it may be useful to know how all of these variables affect the HPA axis or SNS response to the novel cage.

The data are clearly described and well presented, particularly in Supplemental Table 1. In the text of the paper itself, however, it was sometimes unclear the basis on which some conclusions were being drawn, likely because it is unusual to perform statistical analyses across individual experiments. My primary recommendation is to consider a biostatistical collaborator to potentially build a more informative multivariable model. In the absence of this more robust analysis, the strength of some conclusions are questionable and I suggest focusing on the within-experiment comparisons.

NSF is of course just one kind of acute stressor, and only 2 diets were used. Further work will be needed to understand how generalizable these findings might be to other stressors and other foods. This limitation warrants further discussion.

*Reviewer #4:*

In Francois et al., "Assessing effects of stress on feeding behaviors in laboratory mice", the investigators use a novelty suppressed feeding (NSF) assay to investigate variables associated with stress-induced hyperphagia and stress-induced hypophagia. Variables assessed included sex, phase of estrus cycle, duration of social isolation, age, circadian timing, and high fat diet exposure. Performing the assay during the dark (active) phase and performing the assay in HFD-fed mice led to hyperphagic responses. Social isolation induced hypophagia in males but longer social isolation was needed in females to produce the same effect. Factors influencing the variability of response were also investigated.

This is a very interesting topic that would be of interest to the scientific community and the lay public at large since stress affects feeding in the majority of people. The conclusions are interesting, however, I think that many are overstated given the small sample sizes and the lack of discussion as to the magnitude of the responses of the individual study mice. The paper perhaps also reads more like a Methods paper as opposed to a mechanistic paper.

1. My biggest concern with the study as written pertains to the sample sizes, which in general are low. Many of the analyses focus on the variability of the responses (eg hyperphagia vs. hypophagia vs. no change) – the % of the study subjects who responded in one of those three manners. It is risky to base these responses on n's of 8-11 in one set of studies, or even on n's of 15-16 in other sets of studies. Since variability of response was such a major component of the analyses, the n's must be bigger. In comparison, V. Krishnan et al., Cell 2007 used over 700 mice to examine variability in behavioral responses to chronic social defeat stress-while that study focused on molecular determinants of the variability and the current study does not – a larger number of study mice should be included for each of the groups. Without that, I think that looking at the average responses of the entire set of equivalently treated mice is the most that should be reported. A similar problem exists for the estrus cycle data. After separating the larger group of females into those in Diestrus, proestrus, or estrus, then numbers in each group decrease a lot – to n=3 or n-2 in some cases. More mice are needed.

2. Another major concern is the focus only on % of study subjects that respond in one way or another as opposed to the magnitude of the changes. Of those mice that were hyperphagic, what % increase in food intake was observed? This addresses physiologic significance of the results – a 1% average increase is far different than a 200% average increase. Those types of analyses examining the magnitude of the changes for should also be performed/analyzed/discussed. As just one example, on line 184, it is mentioned that 100% of males and 82% of females experienced novel environment stress increased latency – but looking at the figure, the amount of latency in each of the male mice was extremely slight.

3. A third major concern is the nature of the paper. The title suggests this will be a review or a Methods paper. The data contained/studies performed give it the feel of a Methods paper – these are the Methods that must be used in order to study stress-induced hyperphagia or stress-induced hypophagia. I do think that the paper would benefit from the addition of a more mechanistic set of data.

Some other concerns follow:

1. I would like to see an expanded mention (with some details) of other/additional studies in the literature in which stress-induced food intake in mice has been investigated.

2. Some details in the Methods are missing:

a. adults (>10weeks) – that description is too vague. Needs to be more specific about ages of mice used. Preferably, there was not too wide of an age range utilized above 10 weeks of age for these studies.

b. For the animals in studied in their home cages, please specify in the Methods whether bedding changes were performed or changes to the physical home cage were performed during the 2 weeks of acclimation or recovery.

3. Line 275 – can the investigators provide data showing that the 90% caloric restriction was “matched”.

---

## [Author Response]

Essential revisions:1) The authors should assess the question of whether palatability might impact latency of anxiety behavior and/or energy intake.

This is an excellent point. We addressed this issue by adding two cohorts of lean males and females that were only exposed to the palatable high fat diet during the acclimation period and the test itself (hereafter referred to as “acute”) (Figure 5-Supplement 1). The lean and chronic HFD-fed groups exhibited distinct behavioral patterns when presented with HFD in the test. As anticipated, lean mice had >10-fold lower latencies and consumed >10 times more calories of the palatable HFD than the chow diet in the home cage.

In the novel cage, lean mice presented with HFD increase their latency and *decrease* food intake. Other than the fact that we are performing the test at the start of the dark cycle, the acute exposure paradigm is very similar to the “Novelty Induced Hyponeophagia” assay (Dulawa and Hen, 2005 Neurosci Biobehav Rev), which also produces stress-induced suppression of food intake in males. Therefore, novel stress induced increases in food intake seen in chronic HFD exposure is unlikely to be caused by the palatability of the test diet.

2) The authors need to reassess their statistical analyses. It is advisable to recruit a biostatistical collaborator to potentially build a more informative multivariable model. In the absence of this more robust analysis, the strength of some conclusions are questionable and the authors should focus on the within-experiment comparisons.

We followed this excellent advice and recruited a biostatistician, who reanalyzed all of our data using mixed models (Supplemental Tables 1-8). We replaced all simple comparisons between cohorts throughout the manuscript with these analyses.

Importantly, these analyses supported most of our core conclusions. The only change we needed to make was with respect to the effect length and timing of social isolation. We previously claimed that both factors are important for food intake. However, the new analyses revealed that the length of social isolation influences food intake but not latency, while the age at which social isolation is imposed influences latency but not food intake.

3) Another and related concern is the sample sizes. More mice are needed.

The multivariable statistical model showed that social isolation does not interact with estrous cyclicity to influence latency or food intake. This allowed us to combine the two female groups from the old Supplemental Figure 3 into the new Figure 4-Supplement 1 to increase statistical power.

We removed the data segregated by estrous cycle phase where the sample sizes were too small (Supplemental Figure 5), as these were supplemental and not necessary for the main conclusions of the study.

4) Another concern is the focus only on % of study subjects that respond in one way or another as opposed to the magnitude of the changes. This needs to be addressed.5) Comparisons of variance in the feeding response are needed to support the conclusions regarding the effect of a given variable on the likelihood of changing food intake in response to stress.

We removed all of these graphs from the manuscript and eliminated categorical discussion of each experimental outcome in terms of % increased or decreased.

6) The inclusion of other anxiety-like tests is required as fasting modulates anxiolytic effects and this effect increases with the fasting time length (Changhong L et al., 2019; Heinz DE et al., 2021) -- This confounding needs to be addressed.

By performing the assay in the dark phase, we alleviate the need for a fast and increase physiological relevance to humans. The level of caloric restriction in all of our dark cycle assays is only 10%.

7) The use of another stress paradigm would be required to validate the major points of the work.

In humans and rodents*,* males and females experience stress-induced changes in food intake, with sex differences in the impacts of distinct types of stress**.** We acknowledge that our studies capture a single type of stress that may or may not be generalizable to others. We recently published a review in Biological Psychiatry that highlighted the need to develop reproducible and validated assays to study stress-induced feeding behaviors in rodents. This manuscript represents a first step toward this effort. We cited this review both in the Introduction and Discussion and added a sentence in the Summary on the need to repeat these results with other stress paradigms to determine whether these findings are generalizable.

Reviewer #1:Francois and colleagues describe results from a series of studies assessing the effects of stress on feeding behavior in mice. Notably, the authors provided detailed assessment of various parameters while taking into account sex, age, circadian and estrous cycles. The studies are very well described and provide data that will be of wide interest.The focus on anxiety and eating is well justified. The potential link to eating disorders in humans is briefly mentioned. It would be useful if the current results in context of sex differences and eating disorders would be welcome.

Sex differences are most striking in anorexia nervosa. We do not believe that these acute assays can be used to model the chronic and relentless restriction of food intake in this disease. Stress eating, on the other hand, is equally prevalent in males and females in our assays, as well as in humans. This latter point is discussed.

Reviewer #2:This work addresses a range of variables and contradictory findings in analyses of the anxiety-feeding interface by rigorously evaluating several parameters influencing feeding behaviors. While clinical evidence shows susceptibility to either hyperphagia or hypophagia that is predictable and sexually-dimorphic at an individual level, population conclusions have been more difficult to predict. Specifically, factors such as sex and BMI have been implicated in human over- or under-eating under stress; however, a clear picture has not emerged in experimental animal studies due to lack of evaluation and/or standardization of these parameters. Francois et al., systematically evaluate how developmental (and 24h) timing of social isolation impacts feeding in a sex-dependent manner, identify daily timing of novelty-induced feeding behaviors. Their work also probes how diet-induced obesity impacts feeding responses under stress and provide recommendations for creating predictable responses in animals that allow for best practices in studies of stress-induced feeding. Further, they identify that proper design of these experiments can lead to much greater predictability of feeding responses, indicating that the benchmarks herein will enlighten future neurocircuit-level analyses of stress-related appetitive behaviors.While the behavioral characterizations were robustly conducted and provide important new insights to calibrate the entire and broad field of obesity research, a few considerations would enhance the final form of this manuscript. In particular, the question of whether palatability might impact latency of anxiety behavior and/or energy intake, and whether such factors may change according to nutrient prehistory may be helpful (e.g., are DIO mice driven more via rival hedonic vs homeostatic responses).

See the response provided to issue #1.

Is there any way to parse out more within the present work whether the latency phenomenon vs energy intake aspect might reflect differences in "homeostatic drive": for instance, are leptin(or active ghrelin) levels (or leptin-supressed refeeding responses) different amongst males/females, preadolescent/vs. post-pubertal, normal/vs. high-fat fed animals? The point would be to define with endocrine or biochemical quantification the observed distinction between latency and food intake endpoints. Equally exciting would be the observation that ACTH/cortisol levels might vary in a way that could provide a similar entry point for future endocrine analyses (or at least for these points one might review relevant existing literature to provide context from the starting point here).

This was an excellent suggestion. We performed exploratory studies to identify potential neuroendocrine biomarkers of latency and food intake outcomes. We measured ghrelin and CORT in some trunk blood samples that were collected at the end of either the home or novel cage test (randomized). (We did not have enough serum to measure leptin as well.) There are several important caveats to note. First, we were not able to collect blood during the test, because it would have interfered with behavioral assay. The blood was collected at 30-45 min after the test start. Another limitation of these studies is that the serum was not pre-treated in order to detect acyl-ghrelin, so we were only able to measure total ghrelin.

Novel cage stress resulted in higher ghrelin levels in mice with chronic exposure to HFD (Author response image 1 right panel) but not in lean mice (Author response image 1 left and middle panels). This raises the possibility that ghrelin contributes to the hyperphagic response in chronic HFD-fed mice. However, the levels of total serum ghrelin did not correlate with latency or caloric intake.

**Author response image 1. sa2fig1:** 

We elected not to present these data in the manuscript because 1) we were not able to measure acyl-ghrelin; and 2) since ghrelin levels decline rapidly after the consumption of food, it is not clear what are samples are measuring. Genetic or pharmacological manipulations of ghrelin signaling will be needed to define the contribution to stress-induced hyperphagia in chronic HFD-fed mice. This is a focus on our current research efforts.We used a similar strategy in exploratory measurements of CORT. We compared groups of lean mice that exhibited opposite phenotypes – males and females evaluated in the dark phase. The effects of the novel cage on CORT levels were similar between groups, and did not correlate with caloric intake.

Importantly, in all three groups, CORT levels were not significantly increased because several individuals showed levels of CORT similar to home cage levels. This likely reflects the fact that mice were euthanized 30 to 45min after the start of the stress to observe the effects on food intake, and not at the start of the assay where the effects of stress are maximal.

To capture the true effect of novel cage stress on ghrelin and CORT levels, blood sampling should be performed in a separate cohort that is euthanized closer to the beginning of the assay. However, early sampling would then preclude the use of these cohorts to study behavior.

– Addition of more behavioral assays or discussion of such an approach may extend context for the NSF paradigm.

See response to issue #7.

– Question: With regard to social isolation and 24h timing in males vs females, is it possible that females are more synchronized when group-housed--would difference in estrus days explain variance within female experiments?

We did not report 24h timing, nor behavior in group-housed mice. When performed in the Biodaq, mice showed similar behavior regardless of their estrous cycle phase. In addition, when the test was performed in the dark, female behavior was now more consistent than males, despite the fact that caloric intake was impacted by estrous cyclicity in the home cage. Altogether, this suggests that the variance within female is not driven by variation in sex hormones when the assays are performed in the Biodaq environment.

– As presented the work analyzed animals under light:dark conditions, so in formal terms the observations describe "diurnal" aspects of behavior rather than "circadian"-the latter requiring analyses under constant conditions such 24h dark:dark (a semantic point that may be a consideration to those in the field in the future).

Thank you for this clarification. We replaced “circadian” by “diurnal” throughout the paper.

Reviewer #3:In this report, Marie Francois and colleagues set out to uncover aspects of experimental paradigms that promote hyperphagic vs hypophagic responses to psychological stress in mice. The comprehensive experiments were built upon the novelty-suppressed feeding (NSF) test, that is commonly used as an index of anxiety-like behavior. The authors make the salient point that collecting caloric intake data in this paradigm could expand the information revealed by the assay and potentially provide a better understanding of how, and under what experimental conditions, acute stress increases vs decreases caloric intake in mice. They performed 13 variations of the NSF test, to determine the influence of sex, estrous cycle, time of day, age and duration of social isolation, and diet on anxiety like and feeding behavior.The data strongly support that latency to eat in the novel environment, an index of anxiety, was robustly increased across experimental conditions and in both sexes. By contrast, effects on feeding behavior were more variable. Most convincingly, in Figure 4, the authors demonstrate that females socially isolated for ~6 weeks beginning in adolescence exhibit novelty-suppressed caloric intake whereas females socially isolated for 2 weeks beginning in adulthood exhibit novelty-exaggerated caloric intake, when the test is performed during the normal feeding period at the onset of dark. Although the relative contributions of length of isolation and age of isolation onset remain entangled, these findings point towards two key variables that may influence the well-characterized divergent response to stress-eating in women setting the stage for mechanistic studies. Another factor that may influence the direction of the feeding response to stress is obesity, since DIO male and female mice exhibited a hyperphagic response to the novel environment. This conclusion could be strengthened by a direct comparison between lean and obese mice in the same experiment.Strengths:– The important research question and creative approach.– Careful attention to rigor in experimental design including counterbalancing the home cage and novel cage tests, training mice to eat under the test conditions, measuring caloric intake for 30 minutes from the first bite to avoid confounding latency and caloric intake, attention to estrus cycle as a potential variable, and attention to degree of caloric restriction as a potential variable.– Use of the automated BIODAQ feeding system represents an important innovation allowing for feeding to be assessed under more ethologically-relevant conditions.– Clear and easy to digest figures and methods.Weaknesses:The primary concern is regarding the statistical design and analysis.– Essentially, the authors have created a powerful, large multifactorial experimental design but there is a missed opportunity to extract robust information since analyses are not integrated in a single model. Rather, there are mostly a series of pairwise comparisons, some of which span experiments.

We appreciate the constructive suggestion. See the response to issue #2.

– By my understanding, the data were collected across ~10 individual experiments represented by the various figures, and presumably within each experiment the groups were assayed simultaneously (e.g., in Figure 4 females isolated in adolescence and females isolated in adulthood were tested in the NSF test at the same time). However, many of the statistical comparisons shown in Supplemental Table 1 and resulting conclusions occur across experiments. There is less confidence in these conclusions/ findings since behavioral responses to stress are extremely labile and easily influenced by stochastic environmental conditions, which are not controlled when comparing across experiments.

With the help of a biostatistician, we replaced simple statistical comparisons across experiments with a multivariable statistical analysis of our data that controlled for confounding factors such as age, body weight, and order of the tests (home first or novel first).

– Comparisons of variance in the feeding response are critical to several conclusions regarding the effect of a given variable on the 'likelihood of hyperphagic vs hypophagic responses to stress'. However these comparisons are generally only narrative, rather than employing statistical tests to compare variance (or likelihood) between groups, and they also rely on binning the change in caloric intake to categorical variables rather than presenting the data as a continuous variable.

See the response to issues #4 and 5.

Additional physiological endpoints could begin to inform mechanism. For example, how the various timing(s) of social isolation affect body weight and whether weight is a significant overall predictor of the feeding response to stress. Likewise, it may be useful to know how all of these variables affect the HPA axis or SNS response to the novel cage.

We added a sentence in the results stating that body weights were similar between groups that were submitted to different social isolation length and timing. Moreover, weight and feeding responses to stress were not correlated (added to Figure 5F). Body weight was also included as a potential confounding factor in the multivariable analysis.

CORT responses were similar between males, females socially isolated in adulthood, and females socially isolated during adolescence, as shown in the response to Reviewer #1. Importantly, in all three groups, CORT levels were not significantly increased in the novel cage because several individuals showed levels of CORT similar to home cage levels. This could reflect the fact that mice were euthanized 30 to 45min after the start of the stress to observe the effects on food intake, and not at the start of the assay where the effects of stress are maximal.

To capture the true effect of novel cage stress on ghrelin and CORT levels, blood sampling should be performed in a separate cohort that is euthanized closer to the beginning of the assay. However, early sampling would then preclude the use of these cohorts to study behavior. Exploring the contributions of neuroendocrine factors, such as ghrelin, CORT and leptin, to our stress eating model is a focus of our current research efforts.

The data are clearly described and well presented, particularly in Supplemental Table 1. In the text of the paper itself, however, it was sometimes unclear the basis on which some conclusions were being drawn, likely because it is unusual to perform statistical analyses across individual experiments. My primary recommendation is to consider a biostatistical collaborator to potentially build a more informative multivariable model. In the absence of this more robust analysis, the strength of some conclusions are questionable and I suggest focusing on the within-experiment comparisons.

We appreciate the constructive suggestion. See the response to issue #2.

NSF is of course just one kind of acute stressor, and only 2 diets were used. Further work will be needed to understand how generalizable these findings might be to other stressors and other foods. This limitation warrants further discussion.

We appreciate the constructive suggestion. For our discussion of other stressors, see the response to issue #7.

The lack of standardization in test diets was an issue we highlighted in our review of assays involving stress and feeding behavior that was recently published in Biological Psychiatry. We added a sentence in the Summary paragraph mentioning the need to repeat these results with other stress paradigms and with other diets.

Reviewer #4:In Francois et al., "Assessing effects of stress on feeding behaviors in laboratory mice", the investigators use a novelty suppressed feeding (NSF) assay to investigate variables associated with stress-induced hyperphagia and stress-induced hypophagia. Variables assessed included sex, phase of estrus cycle, duration of social isolation, age, circadian timing, and high fat diet exposure. Performing the assay during the dark (active) phase and performing the assay in HFD-fed mice led to hyperphagic responses. Social isolation induced hypophagia in males but longer social isolation was needed in females to produce the same effect. Factors influencing the variability of response were also investigated.This is a very interesting topic that would be of interest to the scientific community and the lay public at large since stress affects feeding in the majority of people. The conclusions are interesting, however, I think that many are overstated given the small sample sizes and the lack of discussion as to the magnitude of the responses of the individual study mice. The paper perhaps also reads more like a Methods paper as opposed to a mechanistic paper.1. My biggest concern with the study as written pertains to the sample sizes, which in general are low. Many of the analyses focus on the variability of the responses (eg hyperphagia vs. hypophagia vs. no change) – the % of the study subjects who responded in one of those three manners. It is risky to base these responses on n's of 8-11 in one set of studies, or even on n's of 15-16 in other sets of studies. Since variability of response was such a major component of the analyses, the n's must be bigger. In comparison, V. Krishnan et al., Cell 2007 used over 700 mice to examine variability in behavioral responses to chronic social defeat stress-while that study focused on molecular determinants of the variability and the current study does not – a larger number of study mice should be included for each of the groups. Without that, I think that looking at the average responses of the entire set of equivalently treated mice is the most that should be reported. A similar problem exists for the estrus cycle data. After separating the larger group of females into those in Diestrus, proestrus, or estrus, then numbers in each group decrease a lot – to n=3 or n-2 in some cases. More mice are needed.

We appreciate the constructive suggestion. See the responses to issue #2-5.

2. Another major concern is the focus only on % of study subjects that respond in one way or another as opposed to the magnitude of the changes. Of those mice that were hyperphagic, what % increase in food intake was observed? This addresses physiologic significance of the results – a 1% average increase is far different than a 200% average increase. Those types of analyses examining the magnitude of the changes for should also be performed/analyzed/discussed. As just one example, on line 184, it is mentioned that 100% of males and 82% of females experienced novel environment stress increased latency – but looking at the figure, the amount of latency in each of the male mice was extremely slight.

See the response to issues #4 and 5.

3. A third major concern is the nature of the paper. The title suggests this will be a review or a Methods paper. The data contained/studies performed give it the feel of a Methods paper – these are the Methods that must be used in order to study stress-induced hyperphagia or stress-induced hypophagia. I do think that the paper would benefit from the addition of a more mechanistic set of data.

We submitted the manuscript as a “Tools and Resources” paper, which has the “potential to substantially accelerate discovery in any area of biology or medicine.” Our studies will broadly enhance efforts to study stress eating. We comply with the requirement to “fully describe the tool or resource so that prospective users have all the information needed to deploy it within their own work.”

Some other concerns follow:1. I would like to see an expanded mention (with some details) of other/additional studies in the literature in which stress-induced food intake in mice has been investigated.

We recently published a review in Biological Psychiatry that discusses how various parameters impact the effects of stress on food intake in animal models and in humans. We now cite this review in relevant sections of the revised manuscript.

2. Some details in the Methods are missing:a. adults (>10weeks) – that description is too vague. Needs to be more specific about ages of mice used. Preferably, there was not too wide of an age range utilized above 10 weeks of age for these studies.

We added more details in the methods section regarding the range of ages used in this study (12 to 30 wk-old). Difference in ages can be explained by the fact that mice were bred in house, and space in the Biodaq was a limiting factor (only 16 cages for a total of 197 mice included in the present manuscript). However, ages were counterbalanced between cohorts and multivariable analysis controlled for age as a potential confounding factor.

b. For the animals in studied in their home cages, please specify in the Methods whether bedding changes were performed or changes to the physical home cage were performed during the 2 weeks of acclimation or recovery.

We added a sentence in the methods specifying that the bedding was left unchanged throughout the ~2 weeks of the paradigm.

3. Line 275 – can the investigators provide data showing that the 90% caloric restriction was "matched".

We added a graph in Figure 4- Supplement 2B describing how 1hr food restriction at night achieves the same caloric intake as a 12hr food restriction during an overnight fast.